# PART-X-MLLM: PART-AWARE 3D MULTIMODAL LARGE LANGUAGE MODEL

**Chunshi Wang**[*,1,2], **Junliang Ye**[‡,*,2,3], **Yunhan Yang**[*,2,4], **Yang Li**[2], **Zizhuo Lin**[1]
**Jun Zhu**[3], **Zhuo Chen**[2], **Yawei Luo**[1,†], **Chunchao Guo**[2,†]
[1]Zhejiang University, [2]Tencent Hunyuan, [3]Tsinghua University, [4]The University of Hong Kong

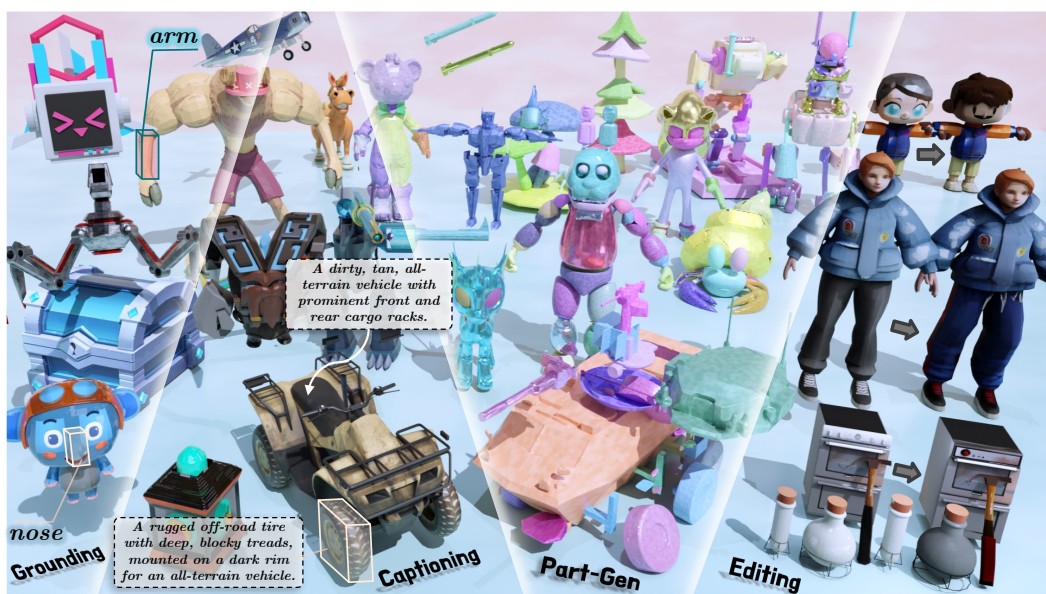

Figure 1: Part-X-MLLM is a natively 3D, part-aware multimodal large language model that provides comprehensive understanding of 3D shapes and supports a wide range of 3D understanding tasks. It also seamlessly integrates with diffusion-based pipelines, enabling semantically precise part-aware 3D shape generation and editing.

## ABSTRACT

We introduce Part-X-MLLM, a native 3D multimodal large language model that unifies diverse 3D tasks by formulating them as programs in a structured, executable grammar. Given an RGB point cloud and a natural language prompt, our model autoregressively generates a single, coherent token sequence encoding part-level bounding boxes, semantic descriptions, and edit commands. This structured output serves as a versatile interface to drive downstream geometry-aware modules for part-based generation and editing. By decoupling the symbolic planning from the geometric synthesis, our approach allows any compatible geometry engine to be controlled through a single, language-native frontend. We pre-train a dual-encoder architecture to disentangle structure from semantics and instruction-tune the model on a large-scale, part-centric dataset. Experiments demonstrate that our model excels at producing high-quality, structured plans, enabling state-of-the-art performance in grounded Q&A, compositional generation, and localized editing through one unified interface. Project page: https://chunshi.wang/Part-X-MLLM/

---

[*]Equal contribution. [†]Corresponding Author. [‡] Project Leader.

## 1 INTRODUCTION

The creation of rich, interactive 3D worlds is a cornerstone of modern visual computing. While recent advances in generative AI have solved the creation of holistic 3D shapes, they largely treat assets as static, monolithic forms. This results in a fundamental limitation we term "structural opaqueness"—where the model perceives a 3D object as a single, indivisible block of geometry rather than a collection of distinct components. Such opaqueness prevents downstream applications from accessing or manipulating specific parts (e.g., editing just "the chair's left leg"), thereby hindering fine-grained control in animation and editing. Real-world objects are inherently assemblies of meaningful parts. Unlocking true 3D interaction, therefore, demands a native LLM-based interface capable of reasoning about this substructure. Unlike approaches that rely on external adapters, our model adopts a native strategy by treating 3D structure as an intrinsic part of its language—processing geometric parts and edit commands as native tokens alongside natural text.

Current 3D Multimodal Large Models (MLLMs) fall short of this goal. Scene-level 3D MLLMs align point clouds with language and perform captioning or Q&A Xu et al. (2024); Hong et al. (2023); Qi et al. (2024b;a), but they largely treat objects as monolithic and lack persistent part identifiers, grounded references, and executable outputs. On the generative side, geometry-oriented models offer high-fidelity asset synthesis via structured 3D latents Xiang et al. (2024); Zhao et al. (2025c); Hunyuan3D et al. (2025b) or tokenized 3D representations Wang et al. (2024); Ye et al. (2025a), yet expose limited semantic addressability. Part pipelines either lift 2D segmentations to 3D Liu et al. (2024a); Chen et al. (2025a); Yang et al. (2024b); Liu et al. (2025a); Yang et al. (2025a)—prone to view inconsistencies and weak 3D constraints—or generate parts natively in 3D Chen et al. (2025b); Zhang et al. (2025); Yang et al. (2025b) without a unified language interface. Editing methods increasingly operate in 3D space Li et al. (2025), but are not themselves language-native frontends. There is still no model that (i) understands and names parts, (ii) grounds references to persistent bounding box (BBox), and (iii) compiles executable add/delete/modify programs while delegating to strong geometry engines—with controllable semantic granularity (from coarse labels to fine descriptions)—through a single instruction-following interface.

We address this challenge with **Part-X-MLLM**, a native 3D part-aware Multimodal Large Language Model that reframes 3D interaction as a language modeling problem. Our core insight is that a spectrum of disparate tasks—generation, editing, and question answering—can be unified under a single, geometry-aware grammar of parts. Part-X-MLLM translates user instructions and 3D visual input into a structured program, emitting a single token sequence of part-level bounding boxes, persistent references, semantic descriptions, and edit operators. This discrete, language-native interface provides three concrete benefits. (1) **Stable part identity and grounding:** tokens carry persistent references to parts via BBox symbols, enabling precise, auditable reasoning and manipulation across steps and tasks. (2) **Controllable semantic granularity:** the same program can surface either coarse labels or fine descriptions on demand, and our post-hoc clustering supports user-controlled merging of parts. (3) **Separation of structure and semantics:** a dual-encoder design decouples geometry (XYZ+normals) from appearance (RGB), avoiding the representational conflict observed in single-encoder ablations and yielding consistent gains on box listing, multi-part grounding, and part Q&A. Because the output program is model-agnostic, any geometry module can be driven by this token interface—turning language into a universal control surface for 3D assets. Empirically, the resulting plans enable strong part grounding, compositional generation, and localized editing across 11 task families on our **UniPart-Bench**, establishing a general paradigm for part-centric 3D intelligence.

Our contributions are summarized as follows:

- We introduce **Part-X-MLLM**, a native 3D part-aware MLLM that unifies generation, editing, and reasoning as a single *geometry-aware program* in a part grammar with persistent BBox tokens—providing a language-native, model-agnostic control surface for 3D assets.

- We propose a **dual-encoder** architecture that decouples structure (XYZ+normals) from appearance (RGB), avoiding representational conflicts and delivering consistent gains over a single-encoder baseline across grounding, captioning, and part Q&A.

- We enable **semantic granularity control** by clustering part bounding boxes using text semantics, allowing seamless transition between coarse components and fine-grained parts under the same programmatic interface.

- We establish **UniPart-Bench**, a 30k-entry part-centric benchmark spanning 11 task families with geometric and linguistic metrics, and use it to rigorously evaluate plan quality and downstream performance.

## 2 RELATED WORK

### 2.1 3D MULTIMODAL UNDERSTANDING AND GENERATION

Early 3D MLLMs align point clouds with language for 3D captioning, QA, and reasoning, including PointLLM Xu et al. (2024), 3D-LLM Hong et al. (2023), Point-BERT Yu et al. (2022), GPT4Point Qi et al. (2024b), and ShapeLLM Qi et al. (2024a). However, point clouds' sparsity and limited detail constrain high-fidelity, editable asset creation. Recent work addresses this through geometry-aware latents: TRELLIS Xiang et al. (2024) employs structured sparse voxel latents with rectified flow for unified decoding to meshes/NeRF/3DGS. Hunyuan3D 2.x Zhao et al. (2025c); Hunyuan3D et al. (2025b) provides a production-ready pipeline with PBR materials. Discretization enables autoregression: LLaMA-Mesh Wang et al. (2024) feeds OBJ text to LLMs but ignores mesh topology, DeepMesh Zhao et al. (2025a) addresses this with auto-regressive artist-mesh creation enhanced by reinforcement learning, while ShapeLLM-Omni Ye et al. (2025a) compresses 3D into discrete tokens for unified text/image/3D understanding and generation. Despite these advances, most systems remain object- or scene-level Wang et al. (2025); Min et al. (2024); Miao et al. (2025b): Existing methods often lack persistent part identities, grounded references, and executable outputs for downstream geometry engines. We address this by introducing a language-native interface that outputs tokenized bounding boxes and edit programs, enabling part-aware and high-fidelity generation and editing.

### 2.2 PART GENERATION

2D-driven pipelines extract multi-view cues then lift to 3D: Part123 Liu et al. (2024a) and PhyCAGE Yan et al. (2024b) uses SAM Kirillov et al. (2023) masks, PartGen Chen et al. (2025a) segments/inpaints with inconsistency issues, SAMPart3D Yang et al. (2024b) and PartField Liu et al. (2025a) distill priors, and HoloPart Yang et al. (2025a) completes parts with diffusion. These methods suffer from weak 3D constraints. Direct 3D approaches include: PASTA Li et al. (2024a) for primitive composition, AutoPartGen Chen et al. (2025b) for autoregressive generation, Part-Packer Tang et al. (2025) and Frankenstein Yan et al. (2024a) for efficient part representation with constrained space usage, BANG Zhang et al. (2025) for exploded views, and Assembler Zhao et al. (2025b) for assembly sampling. OmniPart Yang et al. (2025b) unifies these approaches via autoregressive box planning followed by TRELLIS-based synthesis. X-Part Yan et al. (2025) scale up vecset-based part generation conditioned on semantics provided by Ma et al. (2025).

### 2.3 3D EDITING

Optimization-based editing utilizes SDS: DreamFusion Poole et al. (2022) and DreamReward Ye et al. (2024) enables text-to-3D generation, Vox-E Sella et al. (2023) adds volumetric regularization, and Instruct-NeRF2NeRF Haque et al. (2023) edits multi-views using InstructPix2Pix Brooks et al. (2023) while optimizing NeRF Mildenhall et al. (2021), and $2^2$DEditor Miao et al. (2025a) extends text-driven editing to volumetric video. Faster alternatives include: Shap-Editor Chen et al. (2024b) for feed-forward latent editing, MVEdit Chen et al. (2024a) as a training-free 3D adapter, and PrEditor3D Erkoç et al. (2025) using DDPM inversion with 2D-to-3D lifting. FocalDreamer Li et al. (2024b) enables part-wise assembly, VoxHammer Li et al. (2025) performs training-free latent editing, NANO3D Ye et al. (2025b) achieves precise and coherent 3D editing without requiring masks in a training-free manner, and Make-Your-3D Liu et al. (2024b) customizes subjects via model co-evolution. Yet these methods are typically tool-side: they do not provide a language-native model that reasons about parts and emits executable edit programs with precise spatial grounding. We target this gap by coupling a part-aware planning interface with strong geometry backends.

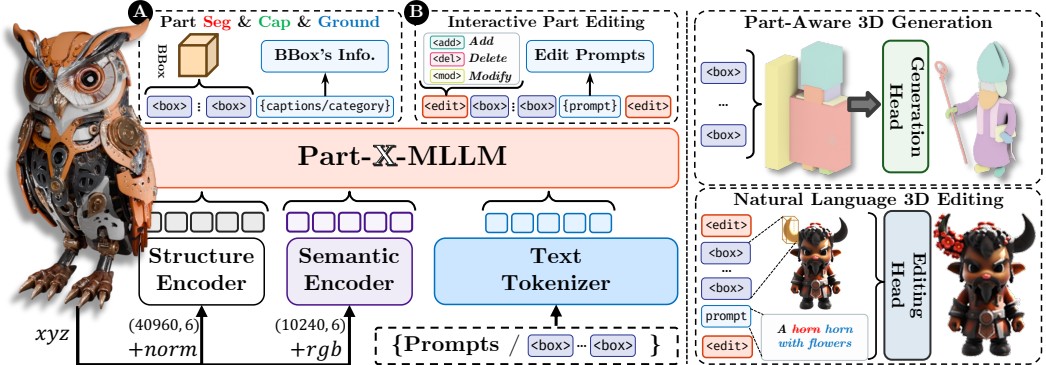

Figure 2: **The Part-X-MLLM Framework.** Our pipeline begins by encoding geometry and appearance features separately using a dual-encoder architecture, which are then fused together with text prompts. These combined features are passed to an autoregressive decoder that generates a program-like token sequence representing a plan (e.g., bounding boxes, edit commands). Finally, specialized geometry heads execute this plan to enable part-aware generation and editing.

## 3 METHODOLOGY

An overview of our framework is shown in Figure 2. Our methodology centers on three key design choices: a unified architecture that processes geometry and language, a multi-stage training curriculum that systematically builds model capabilities, and the use of powerful, pre-existing geometry engines as execution backends.

### 3.1 MOTIVATION

Modern 3D applications demand more than holistic shape synthesis—they require precise, language-driven control over semantically meaningful parts. For example, artists want to swap handles without touching the body; roboticists need to reason about graspable subcomponents; and downstream pipelines rely on consistent, addressable structure for animation and simulation. Prior systems either focus on scene-level understanding or provide powerful but siloed generators/editors with bespoke interfaces. Our goal is a native, part-centric MLLM that treats parts as first-class citizens and exposes a single, executable interface that is intuitive, auditable, and robust across categories.

### 3.2 UNIFIED ARCHITECTURE FOR PART-AWARE PLANNING

**Dual 3D Encoders.** To capture both geometric structure and visual appearance, we employ a dual-pathway encoder. A **Structure Encoder** processes the raw point cloud geometry (XYZ and normals) to extract structural tokens. A parallel **Semantic Encoder** processes RGB color information to produce appearance tokens. This dual representation allows the model to disambiguate parts that may be structurally similar but visually distinct (e.g., two identical chair legs of different colors).

**Structured Planning Language and Autoregressive Decoder.** A decoder-only transformer, initialized from a pretrained LLM, takes the fused sequence of structural, semantic, and text tokens as input. It is trained to autoregressively generate a program-like output that follows our structured planning language. This language defines special tokens for part representation (e.g., <boxs> and <boxe>, representing box-start and box-end, wrapping six quantized coordinate tokens) and edit operations (e.g., <adds>, <dels>, <mods>). By formulating the output as a program, we unify diverse tasks into a single instruction-following problem, where the model's goal is always to generate the correct token sequence representing the plan.

### 3.3 DOWNSTREAM GEOMETRY INTERFACES

Our model's structured output is designed to be consumed by downstream modules capable of interpreting its geometric and semantic content.

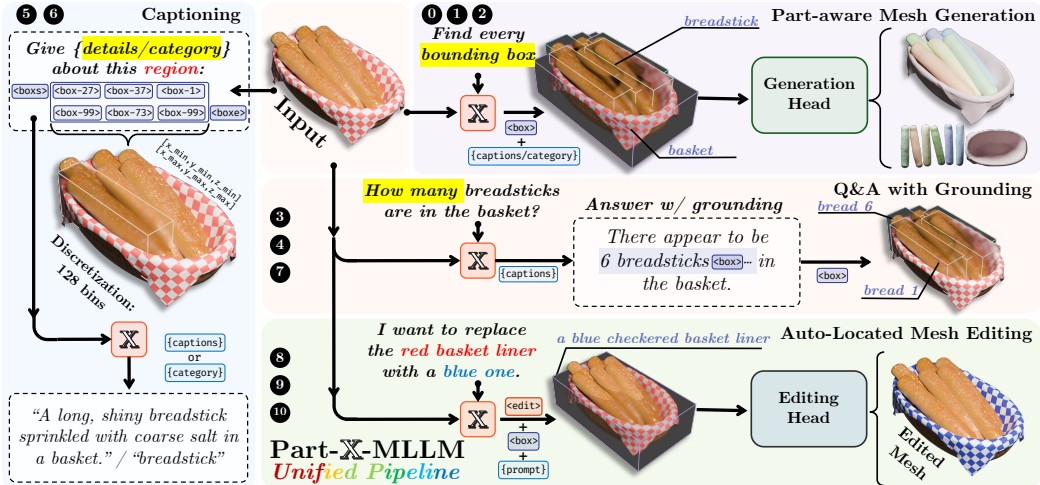

Figure 3: **Task realization with a planning language.** A decoder outputs program tokens that unify diverse interactions: (Top) part-aware generation guided by bounding boxes; (Middle) grounded Q&A whose answers embed BBox tokens; (Bottom) auto-located 3D editing executed via cuboid masks and commands. The numbered circles (e.g., ⊗) denote the corresponding task types.

**Part-Aware Synthesis.** For generation, the planned bounding boxes and optional part text are passed to a synthesis module, which treats the boxes as spatial guides to generate high-fidelity, part-based assets (e.g., in mesh, 3DGS, or NeRF format).

**Localized Editing.** For editing, the emitted program and associated bounding boxes are used to define cuboid masks for localized manipulation, enabling precise edits while preserving untouched regions.

## 3.4 END-TO-END TASK REALIZATION

To make the workflow concrete, Figure 3 illustrates how our structured planning language realizes four representative tasks.

**Part-aware Mesh Generation:** The decoder generates a program containing a set of bounding boxes and optional part text. A synthesis module then uses these boxes as spatial guides to generate a part-based asset. **Q&A with Grounding:** Answers are augmented with BBox tokens, yielding language outputs that carry explicit, persistent references to parts. **Auto-located 3D Editing:** The model localizes the instruction by generating bounding boxes and an edit command (e.g., <adds>). A downstream editing module then uses this program to apply a masked edit.

**Semantic Granularity Control.** Beyond these core tasks, our box-and-text representation enables dynamic control over semantic granularity. By clustering part bounding boxes based on the similarity of their associated text descriptions (using CLIP embeddings), we can progressively merge fine-grained parts into coarser semantic components. This allows users to control the level of detail in the generated output without manual intervention, such as pre-defining the number of parts (cf. PartPacker) or manually merging masks (cf. OmniPart). A qualitative example is shown in Figure 6, with the full algorithm **detailed in the appendix**.

## 3.5 MULTI-STAGE INSTRUCTION TUNING

We adopt a two-stage curriculum. The first stage pretrains a structure-aware encoder for robust geometry understanding. The second stage performs full instruction tuning, integrating a semantic encoder and aligning a powerful LLM with our specialized task grammar.

**Stage 1: Geometry-Only BBox Pretraining.** We initialize the structure encoder with the *Hunyuan 2.1 3D Shape VAE Encoder*. Each training sample is a fixed-size RGB-less point cloud of shape $(40960, 6)$ containing $(x, y, z)$ coordinates and surface normals. The encoder downsamples features by $20\times$ to produce a latent of length 2048. To force bounding-box knowledge into the encoder, we

pair it with a lightweight autoregressive decoder whose task is to predict part-level bounding boxes from these latent features, with no textual semantics involved. After pretraining on 3.6M objects for 10 epochs, we retain the specialized structure encoder weights and discard the lightweight decoder. This stage domain-specializes the 3D encoder to reliably disentangle and localize part BBoxes.

**Stage 2: Full Instruction Tuning with a Dual-Encoder LLM.** After pretraining the structure encoder, we proceed directly to full instruction tuning with a more powerful *Qwen 2.5 VL* model. In this stage, we introduce the *Semantic (RGB) Encoder*, which has the same architecture as the structure encoder and processes a point cloud of shape $(10240, 6)$ with $(x, y, z)$ and $(r, g, b)$ data to capture appearance. We also extend the vocabulary with our task-specific special tokens (e.g., `<boxs>`/`<boxe>`, `<adds>`/`<adde>`). During this stage, we *freeze* the pretrained Structure Encoder from Stage 1 and the *original* Qwen 2.5 VL token embeddings. We then *train only* the new Semantic Encoder, the AR transformer layers of the Qwen 2.5 VL decoder, and the embeddings for our *newly added* special tokens. This approach efficiently aligns the powerful language model with our dual-stream (geometry and appearance) conditioning and executable grammar, preserving its strong prior while adapting it for our specialized tasks.

### 3.6 IMPLEMENTATION AND EXECUTION BACKENDS

To translate plans into high-fidelity geometry, we use powerful, off-the-shelf models as execution backends. For part-aware generation, we use the synthesis module from OmniPart Yang et al. (2025b), feeding it our generated bounding boxes. For editing, we use the training-free volumetric editor VoxHammer Li et al. (2025), providing it with a cuboid mask derived from our planned BBox and the user's instruction. This modular approach allows Part-X-MLLM to serve as a universal, language-driven frontend for various SOTA geometry engines. The rich information encoded in the generated token probabilities also enables advanced downstream tasks, such as **confidence-aware face segmentation** (see Appendix A.5).

## 4 EXPERIMENTS

### 4.1 DATASET

We curate a high-quality, part-centric 3D dataset comprising **85,771** distinct objects with an average of **23** parts per object. Each object is annotated with axis-aligned part bounding boxes (AABBs) and paired natural language annotations at two granularities: a coarse part label (Q1) and a fine-grained part description (Q2). At the object level, we include an overall caption and a small set of instruction–answer pairs for part-aware Q&A. All annotations follow the unified box-token grammar introduced in Section 3, enabling consistent serialization of AABBs and edit programs.

Data construction follows a two-step pipeline: (1) a structured labeling stage collecting object-level and part-level texts and (2) a data building stage converting annotations into instruction-following samples across multiple task families (grounding, captioning, QA, editing). Concretely, we instantiate eleven task templates (Types 0–10) covering pure box listing, multi-part grounding with coarse/fine text, single-part grounding from name or description, box-to-text captioning, part-aware Q&A, and edit programs for deletion/modification/addition. The train/test split is obtained by deterministic file list partition ($\approx 99.5/0.5$). Full details, prompt templates, sampling rules, and dataset statistics are provided in the supplementary material (Tables 10 and figures therein).

### 4.2 EVALUATION PROTOCOL

Since existing benchmarks do not test for structured, part-aware, and executable program generation from language, we introduce **UniPart-Bench**, a held-out set of 400 objects, to evaluate our model's core capabilities. Our evaluation focuses on the quality of the structured plans generated by the model, as measured by the accuracy of the predicted BBox layouts. For downstream tasks, the generated plans are passed to external geometry modules. For generation, we forward the BBoxes to a synthesis head; for editing, we provide the instruction and a cuboid mask derived from the planned BBox.

### 4.3 PART-AWARE GENERATION AND EDITING

**Bounding Box Generation.** To evaluate the quality of our structured generation, we report BBox IoU, Voxel Recall, and Voxel IoU. Matching pairs each ground-truth box with its nearest predicted box. As baselines, we include PartField Liu et al. (2025a) by treating the voxel set as a point cloud and extracting a BBox per predicted segment, and the generation model from OmniPart Yang et al. (2025b). Our model consumes RGB point cloud tokens and a text prompt and autoregressively emits an ordered list of bounding boxes following the box grammar of Section 3. For the PartField baseline, we treat voxels derived from the asset as a point cloud and segment them at the ground-truth part count, then compute bounding boxes per segment for comparison.

Table 1: Quantitative results for bounding box generation (%).

| Method | Voxel recall ↑ | Voxel IoU ↑ | Bbox IoU ↑ |
|---|---|---|---|
| PartField Liu et al. (2025a) | 69.65 | 46.04 | 37.33 |
| OmniPart Yang et al. (2025b) | 72.32 | 47.62 | 39.78 |
| Part-X-MLLM (Ours) | **74.11** | **48.74** | **42.55** |

**Qualitative Generation and Editing Results.** Figure 4 visualizes our qualitative shape decomposition results, where our model demonstrates superior performance in generating semantically coherent and geometrically accurate part segmentations. It successfully captures fine-grained details and maintains structural integrity, outperforming baselines that often produce fragmented or inaccurate decompositions. We also evaluate the model's ability to perform localized, language-driven edits. As shown in Figure 5, Part-X-MLLM successfully interprets user instructions to add, remove, or modify specific parts, executing the edits while preserving the rest of the object's structure.

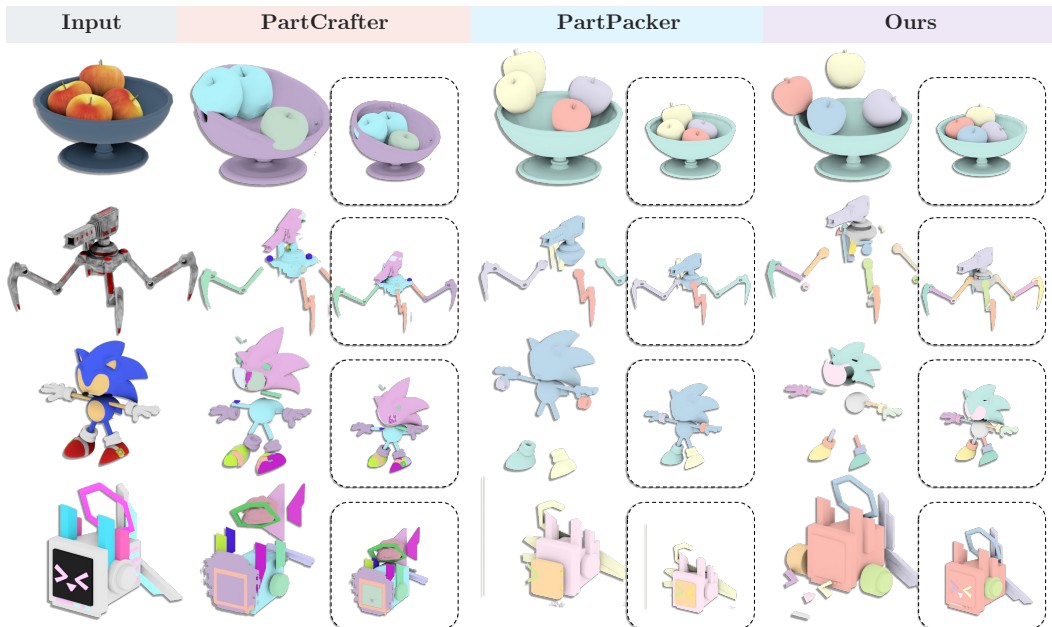

Figure 4: Qualitative shape decomposition results.

**Semantic Granularity Control.** As introduced in Section 3, our framework supports controlling part granularity by semantically clustering bounding boxes. Figure 6 demonstrates this process, where our algorithm progressively merges components based on the CLIP similarity of their textual descriptions, reducing the part count from 22 down to 2. This automated process allows for flexible control over the level of detail without manual intervention.

**Ablation Study: Dual vs. Single Encoder.** We conduct an ablation study to validate our dual-encoder design, which processes geometric structure and visual appearance in separate pathways.

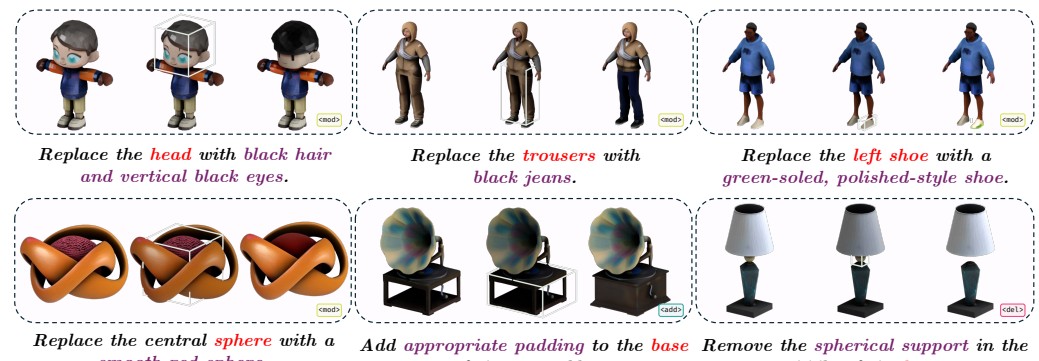

**Figure 5: Qualitative results for part-aware editing.** Our model successfully interprets natural language instructions to perform localized edits, while preserving the integrity of the original object.

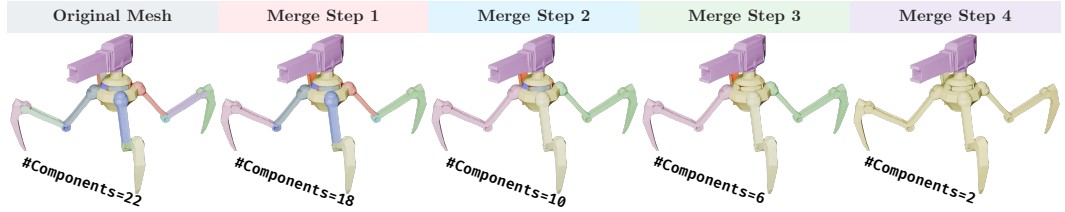

**Figure 6: Semantic granularity control via part clustering.** By clustering parts based on the semantic similarity of their descriptions, we can progressively merge fine-grained components into coarser structures. The number of components is automatically reduced from 22 to 2.

We compare our full model against a single-encoder variant that consumes a unified point cloud with fused geometry (XYZ) and color (RGB) information. As shown in Table 2, the dual-encoder architecture consistently outperforms the single-encoder baseline across all evaluated tasks. For pure geometric tasks like box listing, the dual encoder improves IoU by a significant margin (+7.06). For language-intensive tasks such as Part QA and Multi-Part Grounding, we observe uniform gains across all metrics. This suggests that forcing a single encoder to handle both structural and semantic information creates a conflict, whereas decoupling these responsibilities into two specialized encoders is a more effective and robust design choice.

Table 2: Ablation study on the dual-encoder architecture. We compare our full model against a single-encoder variant. All metrics are reported on **UniPart-Bench**.

| Task | Model | IoU ↑ | SBERT ↑ | SimCSE ↑ | BLEU-1 ↑ | ROUGE-L ↑ | METEOR ↑ |
|------|-------|-------|---------|----------|----------|-----------|----------|
| Pure Box Listing | Dual Encoder (Ours) | **75.53** | - | - | - | - | - |
| | Single Encoder | 68.47 | - | - | - | - | - |
| | △ Gain | *+7.06* | - | - | - | - | - |
| Multi-Part Grounding | Dual Encoder (Ours) | **72.82** | **55.60** | **54.19** | **35.55** | **35.58** | **18.09** |
| | Single Encoder | 69.78 | 54.18 | 53.53 | 33.95 | 33.97 | 17.27 |
| | △ Gain | *+3.04* | *+1.42* | *+0.66* | *+1.60* | *+1.61* | *+0.82* |
| Part QA | Dual Encoder (Ours) | **55.44** | **78.98** | **84.25** | **40.54** | **42.26** | **34.24** |
| | Single Encoder | 54.24 | 78.44 | 83.13 | 39.29 | 41.31 | 33.06 |
| | △ Gain | *+1.20* | *+0.54* | *+1.12* | *+1.25* | *+0.95* | *+1.18* |

### 4.4 PART AND OBJECT UNDERSTANDING

**Part Understanding Q&A.** To evaluate part-level understanding and reasoning, we test on **UniPart-Bench**. We report sentence-level similarities (SBERT, SimCSE) and token-level metrics (BLEU-1, ROUGE-L, METEOR). Results in Table 3 show consistent gains of our method on part-level Q&A. We observe substantial gains over the strongest baseline across all metrics: compared to the best non-ours scores, Part-X-MLLM improves by +17.7 SBERT, +25.8 SimCSE, +17.2 BLEU-1,

+9.7 ROUGE-L, and +9.8 METEOR. These gains reflect stronger part-level grounding and reasoning enabled by our box grammar and instruction tuning.

Table 3: Part understanding Q&A on **UniPart-Bench**.

| Model | SBERT | SimCSE | BLEU-1 | ROUGE-L | METEOR | GPT-5 |
|---|---|---|---|---|---|---|
| GPT4Point Qi et al. (2024b) | 48.32 | 45.17 | 15.16 | 22.55 | 16.19 | 36.99 |
| PointLLM-7B Xu et al. (2024) | 61.30 | 58.48 | 21.78 | 29.26 | 22.45 | 48.68 |
| PointLLM-13B Xu et al. (2024) | 56.36 | 51.47 | 21.40 | 29.16 | 21.80 | 55.83 |
| ShapeLLM-13B Qi et al. (2024a) | 61.19 | 57.26 | 23.32 | 32.56 | 24.45 | 42.21 |
| ShapeLLM-Omni-7B Ye et al. (2025a) | 57.35 | 51.16 | 22.77 | 29.57 | 23.24 | 46.19 |
| MiniGPT-3D Tang et al. (2024) | 58.02 | 53.63 | 21.05 | 28.66 | 22.55 | 50.38 |
| Part-X-MLLM (Ours) | **78.98** | **84.25** | **40.54** | **42.26** | **34.24** | **60.77** |

**Overall 3D Object Captioning.**    Unlike part-level captioning, this benchmark probes holistic object understanding on **UniPart-Bench**. We report SBERT, SimCSE, BLEU-1, ROUGE-L, and METEOR following PointLLM. On overall object captioning, our model also outperforms the best prior scores, with absolute improvements of +4.3 SBERT, +2.5 SimCSE, +18.3 BLEU-1, +19.1 ROUGE-L, and +13.3 METEOR. The large gains on token-based metrics suggest stronger lexical coverage and structure in object-level descriptions.

Table 4: Overall 3D object captioning on **UniPart-Bench**.

| Model | SBERT | SimCSE | BLEU-1 | ROUGE-L | METEOR | GPT-5 |
|---|---|---|---|---|---|---|
| GPT4Point Qi et al. (2024b) | 25.60 | 27.00 | 11.50 | 12.00 | 12.70 | 26.34 |
| PointLLM-7B Xu et al. (2024) | 42.79 | 42.44 | 11.58 | 14.39 | 16.90 | 44.03 |
| PointLLM-13B Xu et al. (2024) | 43.51 | 43.12 | 13.54 | 15.74 | 17.45 | 44.22 |
| ShapeLLM-13B Qi et al. (2024a) | 25.15 | 27.14 | 11.77 | 12.14 | 12.84 | 32.24 |
| ShapeLLM-Omni-7B Ye et al. (2025a) | 31.18 | 31.93 | 17.79 | 19.04 | 14.30 | 30.01 |
| MiniGPT-3D Tang et al. (2024) | 49.52 | 49.44 | 7.75 | 10.23 | 17.24 | 48.75 |
| Part-X-MLLM (Ours) | **53.82** | **51.97** | **36.04** | **38.11** | **30.71** | **55.88** |

**Qualitative Understanding Results.**    Figure 7 provides qualitative examples for overall object captioning. Our model generates more accurate and detailed descriptions compared to baselines. For instance, our model correctly identifies an object as a "pink teddy bear mascot costume with a purple bow tie," while other models provide less specific or incorrect descriptions. Additional qualitative results for part-aware Q&A, demonstrating our model's strong grounding capabilities, are provided in the appendix (Figure 10).

Figure 7: **Qualitative results for overall object captioning.**

## 4.5 SENSITIVITY TO POINT CLOUD RESOLUTION

We evaluate the robustness of Part-X-MLLM by randomly downsampling the input point clouds to various ratios (from 5% to 100%) and measuring performance across Q&A, Captioning, and Part-Level Mesh generation tasks.

Table 5: Sensitivity analysis on input point density.

| Density | Part Q&A | | | | | Overall Caption | | | | | Part-Level Mesh | | |
|---|---|---|---|---|---|---|---|---|---|---|---|---|---|
| | SBERT | SimCSE | B-1 | R-L | MET | SBERT | SimCSE | B-1 | R-L | MET | CD ↓ | F-0.1 ↑ | F-0.05 ↑ |
| 5% | 51.83 | 54.26 | 27.38 | 29.92 | 22.85 | 29.78 | 24.87 | 15.06 | 17.30 | 8.49 | 0.2590 | 0.3188 | 0.3169 |
| **25%** | 76.83 | 82.80 | 39.44 | 40.35 | 32.76 | 52.44 | 48.70 | 36.42 | 38.22 | 30.81 | 0.2287 | 0.6493 | 0.5640 |
| 50% | 78.83 | 83.93 | 39.27 | 40.75 | 32.66 | 54.04 | 50.71 | 37.43 | 39.78 | 31.21 | 0.2318 | 0.6489 | 0.5647 |
| 75% | 78.90 | 84.09 | 39.90 | 41.51 | 33.45 | 53.93 | 51.34 | 36.74 | 38.94 | 30.96 | 0.2240 | 0.6547 | 0.5671 |
| 100% | 78.98 | 84.25 | 40.54 | 42.26 | 34.24 | 53.82 | 51.97 | 36.04 | 38.11 | 30.71 | 0.2226 | 0.6506 | 0.5671 |

As presented in Table 5, our model demonstrates remarkable robustness to input sparsity. Performance metrics across linguistic understanding and geometric generation tasks remain stable even when the input density is reduced to 25%. This indicates that Part-X-MLLM is capable of extracting and reasoning about critical 3D structures from sparse data, ensuring reliable performance across varying input resolutions.

## 5 CONCLUSION

Part-X-MLLM casts 3D interaction as executable program generation: from RGB point clouds and text it emits a single sequence of part AABBs that geometry engines execute, unifying generation, QA, and localized editing, and improving Voxel Recall/IoU and BBox IoU on UniPart-Bench. Appendix A.3.5 supports controllable granularity.

**Limitations and Future Works.** Longer sequences slow inference; simple compaction and hierarchical grouping mitigate latency. Our confidence-based segmentation from BBoxes remains relatively shallow; incorporating stronger features could improve segmentation quality. Fine-tuning on 3D tasks may reduce the base LLM's general language capabilities. In the future, we plan to scale our native part-based planning capability to full indoor scene synthesis, effectively extending the bounding-box grammar from object parts to room-level furniture layouts.

## 6 ETHICS STATEMENT

This work presents **Part-X-MLLM**, a part-aware 3D multimodal model that outputs executable programs (e.g., tokenized AABBs and edit commands). Training uses a blend of publicly available and professionally sourced 3D assets and annotations, subjected to rigorous quality filtering and license review; we avoid personal or biometric data. The model's outputs are grounded and auditable, and the system is intended for research and creative use. We will provide a public API and online interface with usage guidelines. We acknowledge residual risks such as inherited dataset biases and domain shift and will monitor and update the service accordingly. The authors declare no conflicts of interest.

## 7 REPRODUCIBILITY STATEMENT

We detail the structured planning grammar, architecture, training curriculum, and evaluation protocol to enable replication. We will open-source the model checkpoints and the **UniPart-Bench** introduced in this paper, together with evaluation scripts for BBox IoU and voxel metrics, configuration files, prompts/converters for data construction, and complete training/inference code with seeds. A public API and online interface will also be available for lightweight validation.

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

# A APPENDIX

## A.0 THE USE OF LARGE LANGUAGE MODELS (LLMS)

Large Language Models (LLMs) are used exclusively for minor language editing—such as improving grammar and readability—and not for method design or experimental work. All technical contributions, including the methodology, equations, and results, are solely the work of the authors.

## A.1 IMPLEMENTATION DETAILS

Our framework is implemented based on LLaMA-Factory Zheng et al. (2024) and trained on a cluster of 32 GPUs.

**Data Construction Pipeline.** We curated our training data by aggregating 3D assets from large-scale public repositories, primarily Objaverse Deitke et al. (2022) and Objaverse-XL Deitke et al. (2023). To ensure high visual fidelity, we employed an aesthetic scoring model to filter out low-quality or noise-heavy meshes. Given that raw assets often contain overly fragmented components, we applied the Intersection-over-Union (IoU) merging strategy derived from PartPacker Tang et al. (2025). For semantic annotation, we established a multi-view rendering pipeline to render both the holistic appearance of assets and the details of individual parts. These rendered images, including both the complete object renderings and part-specific renderings, were then processed by Qwen-2.5-VL Bai et al. (2025), which generated high-quality, fine-grained textual descriptions.

**Model Architecture.** We utilize Qwen-2.5-VL-3B Bai et al. (2025) as the core multimodal backbone for instruction following. The system features a dual-encoder design initialized from the pre-trained VAE encoders of Hunyuan3D-2.1 Hunyuan3D et al. (2025a). The Structure Encoder accepts geometric inputs (XYZ coordinates concatenated with surface normals) with a resolution of $N = 40,960$ points, projecting them into a latent sequence of length 2,048. The Semantic Encoder processes appearance inputs (XYZ coordinates and RGB colors) with a resolution of $N = 10,240$ points, projecting them into a latent sequence of length 512. Both encoders align their feature dimensions to the LLM's embedding space via linear projection layers. Input point clouds are normalized to the range $[-1, 1]$ along the longest axis, and bounding box coordinates are quantized into 128 discrete bins.

**Training Protocol.** Stage 1 focuses on adapting the Structure Encoder for precise bounding box localization. We employ the open-source OPT-350M Zhang et al. (2022) as a lightweight autoregressive decoder for this task. The model is trained for 10 epochs with a batch size of 128, using the AdamW Loshchilov & Hutter (2019) optimizer (learning rate $1 \times 10^{-4}$, weight decay $1 \times 10^{-5}$, and 5,000 warmup steps). To enhance robustness against scan imperfections, we apply random rotations ($\pm 15°$) and a "Normal Drop" strategy, where surface normals are masked with a 50% probability. In Stage 2, the Structure Encoder is frozen to preserve the learned geometric priors, while the Semantic Encoder, the Qwen-2.5-VL backbone, and special token embeddings are fine-tuned. This stage runs for 60,000 steps with a global batch size of 128 using DeepSpeed ZeRO-2 Rasley et al. (2020) and `bfloat16` precision. We use AdamW Loshchilov & Hutter (2019) with a cosine learning rate scheduler (peak LR $8 \times 10^{-5}$) and enable sample packing with a maximum sequence length of 5,120 to optimize throughput.

## A.2 TASK COMPARISON

Existing 3D models typically present a trade-off between task breadth and semantic granularity. As summarized in Table 6, understanding-focused MLLMs (e.g., ShapeLLM) lack generative capabilities, while generative models (e.g., ShapeLLM-Omni) often operate at the coarse object level without supporting part-level grounding and editing. In contrast, Part-X-MLLM uniquely combines comprehensive understanding, fine-grained part-level operations, and localized editing capabilities in a unified framework.

Table 6: Comparison of capabilities with state-of-the-art 3D models.

| Method | Understand | | Grounding | | Generation | | Modify |
|---|---|---|---|---|---|---|---|
| | Cap | Q&A | Obj | Part | Obj | Part | Edit |
| *Understanding & Reasoning MLLMs* | | | | | | | |
| PointLLM Xu et al. (2024) | ✓ | ✓ | × | × | × | × | × |
| GPT4Point Qi et al. (2024b) | ✓ | ✓ | × | × | × | × | × |
| ShapeLLM Qi et al. (2024a) | ✓ | ✓ | ✓ | × | × | × | × |
| *Unified / Generative Models* | | | | | | | |
| LLaMA-Mesh Wang et al. (2024) | ✓ | ✓ | × | × | ✓ | × | × |
| Hunyuan3D Zhao et al. (2025c) | × | × | × | × | ✓ | × | × |
| ShapeLLM-Omni Ye et al. (2025a) | ✓ | ✓ | × | × | ✓ | × | ✓ |
| *Part-Based Specialists* | | | | | | | |
| Part123 Liu et al. (2024a) | × | × | × | × | × | ✓ | × |
| OmniPart Yang et al. (2025b) | × | × | × | × | × | ✓ | × |
| VoxHammer Li et al. (2025) | × | × | × | × | × | × | ✓ |
| *Ours* | | | | | | | |
| **Part-X-MLLM** | ✓ | ✓ | ✓ | ✓ | × | ✓ | ✓ |

## A.3 MORE EXPERIMENTAL RESULTS

### A.3.1 HUMAN EVALUATION

While quantitative metrics measure geometric alignment, they do not fully reflect human perception of structural logic and editing intent. To address this, we conducted a user study with 32 participants with background in 3D vision. We randomly sampled 25 generated objects and 25 editing instructions. Participants were asked to rate the results on a Likert scale from 1 (Poor) to 5 (Excellent) across four specific dimensions: **Part Plausibility** evaluates whether the decomposed parts in generation tasks are semantically reasonable and structurally sound (e.g., chair legs are attached to the seat), while **Generation Quality** assesses the overall visual fidelity and completeness of the generated parts. For editing tasks, **Instruction Fidelity** measures whether the operation (add/delete/modify) aligns strictly with the text prompt, and **Editing Quality** evaluates the visual coherence of the edited result, including the preservation of non-edited regions. As shown in Table 7, Part-X-MLLM achieves an average score above 4 across all metrics.

Table 7: **Human Evaluation Results.** We report the Mean Opinion Score (MOS) on a scale of 1 to 5 (higher is better).

| Task Type | Evaluation Metric | Score (1-5) |
|---|---|---|
| **Part Generation** | Structural Plausibility | $4.42 \pm 0.6$ |
| | Generation Quality | $4.25 \pm 0.7$ |
| **Part Editing** | Instruction Fidelity | $4.03 \pm 0.5$ |
| | Editing Quality | $4.31 \pm 0.6$ |

Table 8: Evaluation on the PointLLM Benchmark.

| Model | S-BERT ↑ | SimCSE ↑ | BLEU-1 ↑ | ROUGE-L ↑ | METEOR ↑ |
|---|---|---|---|---|---|
| PointLLM-7B Xu et al. (2024) | 47.47 | 48.55 | 3.87 | 7.30 | 11.92 |
| PointLLM-13B Xu et al. (2024) | 47.91 | 49.12 | 3.83 | 7.23 | 12.26 |
| PointLLM-13B* | 50.15 | 50.83 | **17.09** | **20.99** | **16.45** |
| **Part-X-MLLM (Ours)** | **53.43** | **51.21** | 16.00 | 18.34 | 13.28 |

"*" indicates PointLLM was prompted for shorter captions with no more than 20 words..

### A.3.2 ADDITIONAL EXPERIMENTS ON PUBLIC BENCHMARKS

As shown in Table 8, Part-X-MLLM achieves superior semantic similarity scores, outperforming all baselines. While n-gram metrics are slightly lower than PointLLM-13B*, this reflects our model's shift toward structured, part-aware descriptions. The high semantic scores confirm that Part-X-MLLM maintains factual correctness and demonstrates excellent generalization despite part-centric training.

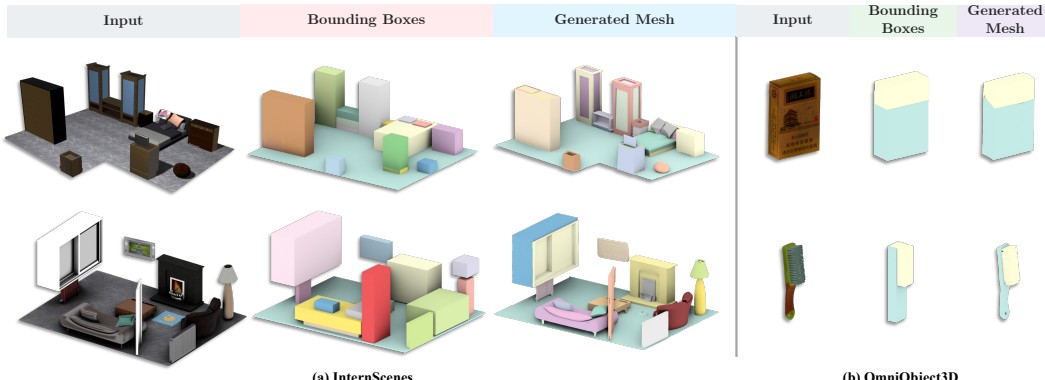

Figure 8: Qualitative evaluation of generalization and robustness.

### A.3.3 GENERALIZATION AND ROBUSTNESS ANALYSIS

Beyond standard object-level synthesis, we further investigate the model's capability to handle out-of-distribution data, as shown in Figure 8. First, we explore scene-level composition using the InternScenes Zhong et al. (2025) dataset. This aligns with the emerging paradigm of compositional scene generation Zhou et al. (2024; 2025); Yang et al. (2024a); Ge et al. (2024); Liu et al. (2025b), where complex environments are constructed from distinct entities. Although trained on object parts, Part-X-MLLM successfully generalizes to this domain by treating individual furniture items as components of a room, generating plausible layouts and meshes in a zero-shot manner. Second, to address the domain gap between synthetic and realistic data, we evaluate the model on real-world scans from OmniObject3D Wu et al. (2023). These inputs typically contain high-frequency noise, holes, and inconsistent normals. Our model demonstrates strong robustness by effectively filtering out these artifacts, producing precise bounding boxes and clean geometric reconstructions even under such challenging conditions.

### A.3.4 ISOLATION OF PLANNING CAPABILITIES

To assess the contribution of our structured planning interface independent of the downstream geometry kernels, we conducted a controlled comparison for the generation task. We compared our method against several baselines, including the native planner of OmniPart Yang et al. (2025b) and pipeline approaches using TRELLIS Xiang et al. (2024) combined with 3D segmentation tools (SAM3D Yang et al. (2023), PartField Liu et al. (2025a), HoloPart Yang et al. (2025a)).

**Setup.** For the "Part-X-MLLM + OmniPart" entry, we use our model to generate the bounding box plan from text, which is then fed into the frozen OmniPart synthesis decoder. This allows for a direct "planner-to-planner" comparison with the original OmniPart method.

As shown in Table 9, when using the exact same generation backend, our planner outperforms the native OmniPart planner in Part-Level metrics (e.g., improving F-0.05 from 0.46 to 0.57). This indicates that Part-X-MLLM produces more geometrically accurate and semantically consistent part layouts (bounding boxes), which in turn enables the backend to synthesize higher-fidelity components.

**Operational Efficiency in Editing.** Beyond quantitative generation quality, our contribution to the editing task lies in a fundamental shift in usability. Native geometry engines like VoxHammer require explicit, manually crafted 3D binary masks to identify edit regions—a process that typically demands **manual modeling** in professional software (e.g., Blender). Part-X-MLLM bridges this

Table 9: Quantitative Comparison of Part-Level and Overall-Level Generation.

| Method | Part-Level | | | Overall-Level | | |
|---|---|---|---|---|---|---|
| | CD ↓ | F-0.1 ↑ | F-0.05 ↑ | CD ↓ | F-0.1 ↑ | F-0.05 ↑ |
| TRELLIS + SAM3D Yang et al. (2023) | 0.58 | 0.25 | 0.20 | 0.11 | 0.89 | 0.72 |
| TRELLIS + PartField Liu et al. (2025a) | 0.24 | 0.60 | 0.42 | 0.11 | 0.89 | 0.72 |
| TRELLIS + PartField + HoloPart Yang et al. (2025a) | 0.24 | 0.61 | 0.43 | 0.09 | 0.90 | 0.74 |
| Part123 Liu et al. (2024a) | 0.47 | 0.28 | 0.14 | 0.42 | 0.36 | 0.20 |
| OmniPart Yang et al. (2025b) | 0.23 | 0.63 | 0.46 | **0.08** | **0.91** | **0.77** |
| **Part-X-MLLM + OmniPart (Ours)** | **0.22** | **0.65** | **0.57** | **0.08** | 0.90 | 0.77 |

gap by acting as an intelligent semantic agent: it translates high-level natural language instructions (e.g., "remove the armrests") directly into precise, geometrically grounded cuboid masks. This **automates the entire workflow**, transforming 3D editing from an expert-only, manual operation into an accessible, fully **language-driven interaction** where users simply "speak" to edit.

### A.3.5 Semantic Part Clustering Algorithm

To enable dynamic control over semantic granularity, we introduce a post-processing algorithm that clusters fine-grained part bounding boxes into coarser, semantically meaningful components. This process, illustrated in Figure 6, operates without requiring manual intervention or a predefined number of target clusters. The algorithm follows a three-step pipeline: feature extraction, clustering, and merging.

**1. Feature Extraction.** For each predicted part $p_i$, we extract its bounding box $b_i = (\mathbf{x}_{\min}, \mathbf{x}_{\max})_i$ and textual description $d_i$. A hybrid feature vector $\mathbf{f}_i$ is then generated.

First, the semantic feature vector $\mathbf{f}_{\text{sem},i}$ is obtained by encoding the description with a pretrained CLIP model:

$$\mathbf{f}_{\text{sem},i} = \text{CLIP-Encode}(d_i). \tag{1}$$

Next, we compute the spatial feature vector $\mathbf{f}_{\text{spat},i}$ from the bounding box's center $\mathbf{c}_i = (\mathbf{x}_{\min} + \mathbf{x}_{\max})/2$ and size $\mathbf{s}_i = \mathbf{x}_{\max} - \mathbf{x}_{\min}$. The raw spatial vector is normalized across all $N$ parts in the object to produce $\hat{\mathbf{f}}_{\text{spat},i}$:

$$\mathbf{f}_{\text{spat},i} = [\mathbf{c}_i, \mathbf{s}_i], \quad \hat{\mathbf{f}}_{\text{spat},i} = \text{Normalize}(\{\mathbf{f}_{\text{spat},j}\}_{j=1}^{N})_i. \tag{2}$$

Finally, the semantic and spatial features are combined using a weighting factor $\alpha \in [0, 1]$, and the resulting vector is L2-normalized:

$$\mathbf{f}_i = \frac{(1-\alpha)\mathbf{f}_{\text{sem},i} \oplus \alpha\hat{\mathbf{f}}_{\text{spat},i}}{\|(1-\alpha)\mathbf{f}_{\text{sem},i} \oplus \alpha\hat{\mathbf{f}}_{\text{spat},i}\|_2}, \tag{3}$$

where $\oplus$ denotes concatenation.

**2. Clustering.** We apply DBSCAN to the set of feature vectors $\{\mathbf{f}_i\}_{i=1}^{N}$. DBSCAN groups points based on two parameters: a distance threshold $\epsilon$ and a minimum number of points 'minPts'. A point $\mathbf{f}_i$ is a *core point* if its $\epsilon$-neighborhood contains at least 'minPts' points. A cluster is formed by a set of *density-connected* points, starting from a core point and recursively expanding to all reachable neighbors. This approach allows us to automatically identify a variable number of clusters $K$ without prior specification, returning a set of clusters $\mathcal{C} = \{C_1, \ldots, C_K\}$ and a set of noise points $\mathcal{N}$.

**3. Merging.** For each cluster $C_k \in \mathcal{C}$, we compute a single merged bounding box $B_k = (\mathbf{X}_{\min,k}, \mathbf{X}_{\max,k})$. This is done by taking the component-wise minimum and maximum over all bounding boxes $b_i \in C_k$:

$$\mathbf{X}_{\min,k} = \min_{i|b_i \in C_k}(\mathbf{x}_{\min,i}), \quad \mathbf{X}_{\max,k} = \max_{i|b_i \in C_k}(\mathbf{x}_{\max,i}). \tag{4}$$

The final output is a set of $K$ merged bounding boxes, representing a coarser, semantically-grouped decomposition of the object.

This automated approach provides a flexible and powerful way to adjust the granularity of the generated 3D assets, bridging the gap between fine-grained part generation and high-level semantic understanding.

## A.4 ADDITIONAL QUALITATIVE RESULTS

Figure 10 provides qualitative examples for part-aware question answering. Our model demonstrates strong grounding capabilities by providing detailed, box-annotated answers that accurately describe object parts in response to user queries.

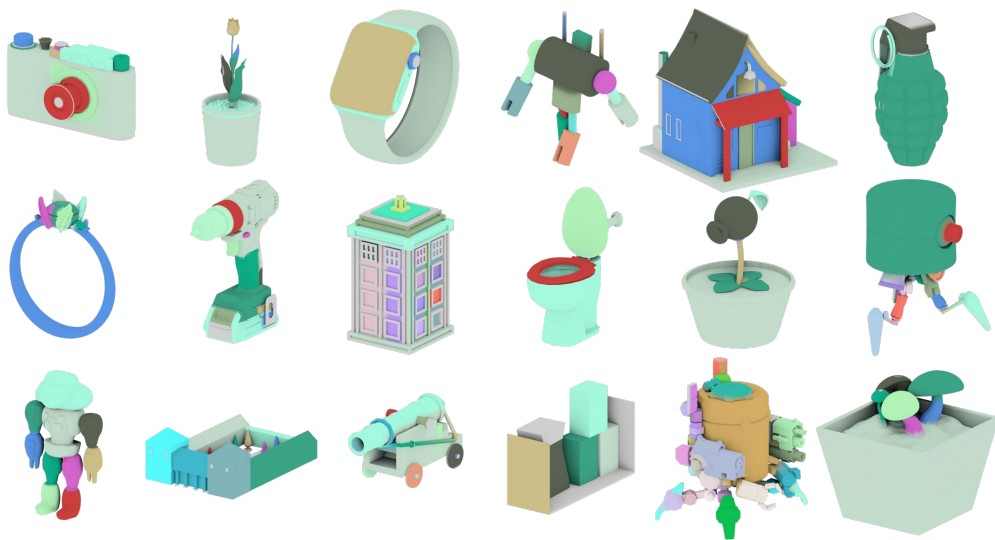

Figure 9: **Confidence-aware face segmentation.** By leveraging the generated bounding boxes and their associated confidence scores, we can achieve high-quality, fine-grained face-level segmentation of 3D objects without any additional training.

## A.5 CONFIDENCE-AWARE FACE SEGMENTATION FROM BOUNDING BOXES

As mentioned in Section 3, the rich information encoded in our model's autoregressive output can be leveraged for advanced downstream tasks beyond simple generation or editing. One such application is fine-grained, confidence-aware face segmentation, as shown in Figure 9. This process requires no additional training and relies solely on the generated bounding boxes and the token probabilities from the decoding process.

The algorithm follows a three-step process:

**1. Confidence-Aware BBox Inference.** During autoregressive decoding, the model generates a sequence of tokens $T = (t_1, t_2, \ldots, t_L)$ that represent a series of bounding boxes. For each token $t_i$, the model also outputs a probability distribution over the entire vocabulary, from which we derive a confidence score. The confidence of a bounding box $B_j$, which is composed of a sequence of $k$ tokens (typically 6), is calculated as the arithmetic mean of the probabilities of its constituent tokens:

$$\text{Conf}(B_j) = \frac{1}{k} \sum_{i=1}^{k} P(t_i | t_{<i}) \tag{5}$$

This provides a per-box confidence score that reflects the model's certainty in its prediction.

**2. Face-to-Box Assignment.** Given a mesh with a set of faces $F = \{f_1, f_2, \ldots, f_M\}$ and a set of inferred bounding boxes $\mathcal{B} = \{B_1, B_2, \ldots, B_N\}$, we first determine which faces belong to which

| Input | GT | PointLLM | Ours |
|---|---|---|---|
| *What is the man wearing?* | *He is wearing a yellow puffer vest*<boxs><box-46><box-73><box-48><box-80><box-117><box-78><boxe> *over a light blue long-sleeved shirt*<boxs><box-48><box-66><box-51><box-78><box-109><box-76><boxe>, *dark blue pants*<boxs><box-49><box-16><box-54><box-77><box-77><boxe>, *and bright green sneakers*<boxs><box-48><box-1><box-55><box-56><box-11><box-75><boxe><boxs><box-70><box-1><box-55><box-78><box-11><box-75><boxe>. | *The man is wearing a yellow shirt and black pants.* | *He is wearing a yellow puffer vest* <boxs><box-45><box-73><box-48><box-81><box-117><box-78><boxe> *over a blue shirt* <boxs><box-48><box-66><box-51><box-78><box-109><box-76><boxe>, *dark blue pants* <boxs><box-49><box-16><box-54><box-77><box-77><boxe>, *and green sneakers* <boxs><box-48><box-1><box-55><box-56><box-11><box-75><boxe> <boxs><box-70><box-1><box-55><box-78><box-11><box-75><boxe>. |
| *What type of firearm is this?* | *This is a modern-style sniper rifle.* | *This is a rifle.* | *This is a bolt-action sniper rifle.* |
| *What is the overall structure of the object?* | *The object consists of a large main casing* <boxs><box-1><box-28><box-25><box-125><box-105><box-92><boxe> *that rests on two long, parallel bars* <boxs><box-5><box-21><box-20><box-18><box-28><box-106><boxe> *and* <boxs><box-100><box-21><box-20><box-112><box-28><box-106><boxe> *which act as its feet.* | *The object appears to be a tall, rectangular structure with a protruding section on one side, a window on the front, and multiple ventilation slits on its sides.* | *The object consists of a main rectangular casing* <boxs><box-1><box-28><box-25><box-125><box-105><box-92><boxe> *with a large circular fan assembly* <boxs><box-12><box-38><box-89><box-68><box-95><box-94><boxe> *and two small feet* <boxs><box-5><box-21><box-20><box-18><box-28><box-106><boxe> *and* <boxs><box-100><box-21><box-20><box-113><box-28><box-106><boxe> *at bottom.* |

*Questions and Answers*

Figure 10: **Qualitative results for part-aware Q&A.** Our model provides more accurate and descriptive answers, with precise part grounding indicated by bounding box tokens.

boxes. A face $f_m$ is considered a candidate for $B_j$ if its centroid $\mathbf{c}_m$ lies within the volume of $B_j$:

$$\mathbf{c}_m \in B_j \iff (\mathbf{c}_m \geq \mathbf{x}_{\min,j}) \wedge (\mathbf{c}_m \leq \mathbf{x}_{\max,j}) \tag{6}$$

where $\mathbf{x}_{\min,j}$ and $\mathbf{x}_{\max,j}$ are the minimum and maximum coordinates of box $B_j$, and the comparison is element-wise.

**3. Conflict Resolution.** A face's centroid may lie within multiple overlapping bounding boxes, creating an ambiguity. We resolve this using a two-tiered rule system:

- **Containment Rule:** If a face $f_m$ is a candidate for two boxes, $B_i$ and $B_j$, and one box is strictly contained within the other (e.g., $B_i \subset B_j$), the face is assigned to the box with the smallest volume. This prioritizes more specific, fine-grained predictions.

- **Confidence Rule:** If the boxes overlap but neither contains the other, the face is assigned to the box with the highest confidence score, $\text{Conf}(B_j)$. This leverages the model's own uncertainty estimate to make the most likely assignment.

This process results in a deterministic assignment of each face to a single bounding box, producing a high-quality, fine-grained segmentation of the object, as shown in Figure 9.

### A.6 ANALYSIS OF SPECIAL TOKEN EMBEDDINGS

To better understand how our model interprets the specialized grammar, we visualize the embeddings of our newly added special tokens using t-SNE, as shown in Figure 11. The visualization reveals a highly structured and semantically meaningful latent space.

We observe three key phenomena. First, the tokens form distinct clusters based on their function: Point, Box, and Edit tokens occupy separate regions of the embedding space. Second, the 128 box tokens, which represent quantized coordinates, form a continuous, ordered manifold. This demonstrates that the model has learned the ordinal nature of spatial coordinates rather than treating them as independent categorical variables. Third, tokens with similar functions, such as the start/end pairs for edits (e.g., `<adds>`/`<adde>`), are positioned closely together. This structured organization confirms that the model has successfully learned a robust and interpretable representation of our executable grammar, which is crucial for precise, language-driven 3D planning.

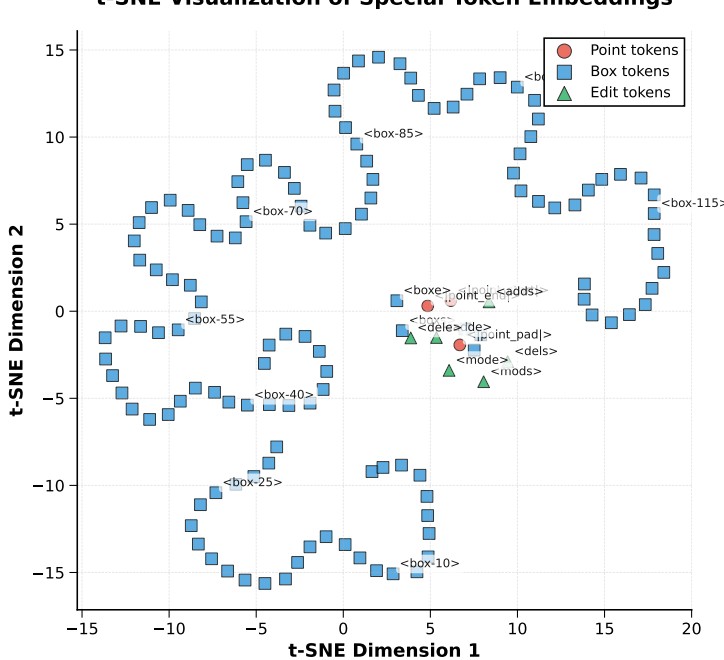

Figure 11: **t-SNE visualization of special token embeddings.** The tokens form distinct, well-structured clusters based on their function, indicating a meaningful learned representation.

Figure 12: Model-assisted labeling pipeline. Left: inputs (full-asset + per-part crops). Middle: structured tool schema drives the LMM to output object-level and part-level JSON. Right: validated JSON is stored and used by the data builder.

## A.7 DATASET CONSTRUCTION AND LABELING

**Scope.** We build a high-quality, part-centric dataset tailored for Part-X-MLLM. The corpus contains **85,771** unique 3D objects with an average of **23** parts per object. Each part is annotated with an axis-aligned bounding box (AABB) and two levels of text: a coarse name (Q1) and a fine-grained description (Q2). At the object level, we include a concise overall caption and a small set of instruction–answer pairs for part-aware Q&A. All annotations are serialized using the unified box-token grammar described in Section 3.

### A.7.1 MODEL-ASSISTED LABELING

To scale high-quality labels consistently, we adopt a model-assisted pipeline guided by a structured tool schema. Given a full-asset render and a sequence of part close-ups, we collect:

- Q1: short part name.
- Q2: fine-grained natural description ($\leq$ 15 words; avoid irrelevant rendering terms).
- Q3: confidence flag (Yes/No).

Concretely, we follow the schema implemented in our labeling tool, which calls an external LMM with a JSON response format and deterministic field ordering. For each object, we provide: (1) one full-asset image (front view); and (2) K part crops (one per part).

### A.7.2 BUILDING INSTRUCTION-FOLLOWING SAMPLES

We convert raw labels into diverse instruction-following pairs covering grounding, captioning, QA, and editing. A central convenience is a *box-token grammar* with opening/closing tokens

---

**Algorithm 1** Data building (simplified)

---

1: Load datas
2: **for** each object $o$ **do**
3:     Serialize each part AABB to tokens; sort by $(z_{min}, y_{min}, x_{min})$
4:     **for** each template $t \in \{0, \ldots, 10\}$ **do**
5:         Instantiate a natural-language prompt from a template pool
6:         Emit the target sequence (boxes, text, or edit program)
7:         Append conversation pair to the corpus
8: Shuffle and save shards; optionally balance per-template counts

---

`<boxs>` and `<boxe>` wrapping six quantized coordinates, and edit verbs `<adds>`/`<adde>`, `<dels>`/`<dele>`, and `<mods>`/`<mode>`.

**Quantization and serialization.** Each coordinate $x \in [-1, 1]$ is quantized into $K = 128$ bins as

$$q(x) = \text{round}\left(\tfrac{x+1}{2}(K-1)\right), \qquad \tilde{x} = 2\,\tfrac{q(x)}{K-1} - 1, \tag{7}$$

then serialized as six tokens inside `<boxs>`...`<boxe>`. For reproducibility, parts in a list are deterministically ordered by $(q(z_{min}), q(y_{min}), q(x_{min}))$.

**Task families.** We instantiate eleven templates (Types 0–10):

- Type 0: pure box listing from a point cloud ("detect all bounding boxes").
- Type 1: multi-part grounding with coarse text (AABBs + Q1 per part).
- Type 2: multi-part grounding with fine text (overall description first, then AABBs + Q2).
- Type 3: single-part grounding from coarse text (locate all Q1 parts; return AABBs + description).
- Type 4: single-part grounding from fine text (locate part by Q2; return a single AABB).
- Type 5: box-to-text (given a box, answer Q1).
- Type 6: box-to-text (given a box, answer Q2).
- Type 7: part-aware QA (replace textual part references `<Part_i>` with the corresponding box tokens in answers).
- Type 8: deletion program (emit `<dels>` [boxes] `<dele>`).
- Type 9: modification program (emit `<mods>` [box] new text `<mode>`).
- Type 10: addition program (emit `<adds>` [box] text `<adde>`).

**Train/test split and balancing.** We partition the file list deterministically at $0.5\%$ for test and $99.5\%$ for train. Templates 1–2 are lightly duplicated to increase multi-part coverage; for templates 3–7 we downsample to a fixed budget; for edit templates (8–10) we cap the number per shard. See Table 10.

## A.8   Dataset Statistics

**Task families and sizes.** Table 10 summarizes per-task counts before/after balancing. Counts follow our build scripts.

**Category distribution.** Our corpus spans everyday objects and scenes. Table 11 lists the main categories (top-12 by frequency).

## A.9   Comprehensive Results on UniPart-Bench

We report per-task results on **UniPart-Bench**. Note that UniPart-Bench is a held-out subset of our 85,771-object training dataset, ensuring identical data construction pipeline and distribution characteristics. Following our data construction, each ground-truth (GT) item may contain both BBox tokens and text. When both are present, we evaluate BBoxes with IoU and text with

Table 10: Task families and sizes. "Raw" denotes counts before optional balancing; "Final" denotes the target budget after balancing.

| Name | Input | Output | Raw | % | Type | Final |
|------|-------|--------|-----|---|------|-------|
| Single-Part Grounding | point + coarse text | 1 box + fine text | 506,755 | 7.30 | T3 | 506,755 |
| Single-Part Grounding | point + fine text | 1 box | 887,590 | 12.78 | T4 | 506,755 |
| Multi-Part Grounding | point + text | all boxes + Q1 | 85,771 | 1.24 | T1 | 257,313 |
| Multi-Part Grounding | point + text | all boxes + Q2 | 85,771 | 1.24 | T2 | 257,313 |
| Box-to-Text (coarse) | point + box + text | Q1 | 887,590 | 12.78 | T5 | 506,755 |
| Box-to-Text (fine) | point + box + text | Q2 | 887,590 | 12.78 | T6 | 506,755 |
| Part QA | point + text | text | 577,369 | 8.31 | T7 | 506,755 |
| Edit—Add | point + text | program (box + text) | 247,998 | 3.57 | T10 | 247,998 |
| Edit—Remove | point + text | program (boxes) | 1,394,345 | 20.08 | T8 | 247,998 |
| Edit—Replace | point + text | program (box + text) | 883,941 | 12.73 | T9 | 247,998 |
| Pure box listing | point + text | all boxes | 500,000 | 7.20 | T0 | 500,000 |
| Total | | | 6,944,720 | 100.00 | | 4,292,395 |

Table 11: Category distribution.

| Rank | Category | Count | Share (%) |
|------|----------|-------|-----------|
| 1 | Human | 20,426 | 23.74 |
| 2 | Industrial goods | 7,139 | 8.30 |
| 3 | Home goods | 7,010 | 8.15 |
| 4 | Buildings | 6,909 | 8.03 |
| 5 | Personal items | 6,730 | 7.82 |
| 6 | Animals | 6,582 | 7.65 |
| 7 | Weapons | 6,406 | 7.45 |
| 8 | Vehicles | 5,996 | 6.97 |
| 9 | Cultural artifacts | 5,995 | 6.97 |
| 10 | Food | 5,885 | 6.84 |
| 11 | Technology & electronics | 5,183 | 6.02 |
| 12 | Others | 1,774 | 2.06 |

SBERT/SimCSE/BLEU-1/ROUGE-L/METEOR. If a GT contains only BBoxes or only text, we evaluate the available modality and leave the other columns blank. Table 12 summarizes results for Tasks 0–10 while mapping each task to its template Type and name as in Table 10.

**Discussion.** Language-intensive tasks (T7 Part QA, T10 Edit—Add) obtain the highest SBERT/SimCSE and strong lexical metrics, indicating robust alignment between our planned box-conditioned answers/programs and textual GT. Among IoU-based tasks, T0/T2/T10 show the strongest geometric alignment, reflecting reliable planning for pure detection, fine grounding, and edit addition respectively. Blank text or IoU entries arise by design when a task's GT lacks the corresponding modality.

## A.10   PROMPT TEMPLATES FOR DATA CONSTRUCTION

To ensure the reproducibility of our dataset construction, this section provides the complete set of English prompt templates used to generate the instruction-following samples for each of the 11 task types, as described in Section A.7.2. These templates are presented in the tables below.

### A.10.1   TYPE 0: PURE BOX LISTING

| ID | Prompt Template |
|----|-----------------|
| 1 | `"Detect all bounding boxes in this point cloud"` |
| 2 | `"Show me all the bounding boxes"` |
| 3 | `"Generate bounding boxes for all objects"` |

| ID | Prompt Template |
|----|-----------------|
| 4 | "Find all object boundaries" |
| 5 | "Extract all bounding boxes from this scene" |
| 6 | "Locate all object bounding boxes" |
| 7 | "Output all detected bounding boxes" |
| 8 | "Provide bounding boxes for all components" |
| 9 | "Identify all object boundaries in this model" |
| 10 | "Return all bounding box coordinates" |
| 11 | "Detect and output all object boxes" |
| 12 | "Find all rectangular boundaries" |
| 13 | "Generate all object bounding boxes" |
| 14 | "Show all detection boxes" |
| 15 | "Output bounding box coordinates for all objects" |
| 16 | "Detect all objects and return their boxes" |
| 17 | "Find every bounding box in this point cloud" |
| 18 | "Extract object boundaries from this 3D data" |
| 19 | "Provide all object detection boxes" |
| 20 | "Return coordinates of all detected objects" |

### A.10.2   TYPE 1: MULTI-PART GROUNDING (COARSE TEXT)

| ID | Prompt Template |
|----|-----------------|
| 1 | "What distinct components does this contain? Please annotate with bounding boxes and provide short labels" |
| 2 | "What functional parts make up this object? First provide 6 box-tokens then write the name" |
| 3 | "What structural elements can be decomposed? Output in the specified format" |
| 4 | "What key components does this have? Please locate and name them" |
| 5 | "What identifiable parts are there? Mark with AABB tokens" |
| 6 | "What construction units can be distinguished? Please list them" |
| 7 | "What parts need to be annotated in this?" |
| 8 | "What basic components does this contain? Please output bounding box + label" |
| 9 | "What main parts is this composed of? Please enumerate using token format" |
| 10 | "What recognizable sub-parts are there? Use the specified format for output" |
| 11 | "Which distinct parts exist here? Provide box-tokens and short labels" |
| 12 | "Identify every component and prepend its 6 quantized box tokens" |
| 13 | "List all separable elements; each line starts with tokens" |
| 14 | "Locate and name each part of the object" |
| 15 | "Enumerate all components with their bounding-box tokens" |
| 16 | "Break the shape into parts, output AABB tokens then a concise tag" |
| 17 | "Mark every structural unit. Format: tokens followed by NAME" |

| ID | Prompt Template |
|---|---|
| 18 | "Point out all functional pieces and give their tokenized boxes" |
| 19 | "Provide the set of parts and their six token indices" |
| 20 | "Give every recognized section together with its AABB tokens" |
| 21 | "List all structural elements using 6 box-tokens + name format" |
| 22 | "Return the quantized bounding box and short name for each part" |
| 23 | "Please enumerate in the format of tokens followed by NAME" |
| 24 | "Output part AABB (tokens) and their names" |
| 25 | "Give the list of components together with their quantized boxes" |
| 26 | "Return each element as six tokens followed by a short label" |
| 27 | "Provide AABB tokens plus name for every distinguishable component" |
| 28 | "Enumerate all parts with their bounding-box tokens and a brief tag" |
| 29 | "Please identify all parts and output bounding box tokens + short name" |
| 30 | "After completion, only return the parts list without extra explanation" |
| 31 | "Output strictly according to the specified format, no additional text" |
| 32 | "No extra description at the end, only list the parts" |
| 33 | "List the token AABB and name for each part" |
| 34 | "Give tokens and labels in order of appearance" |
| 35 | "Use six tokens followed by space and name" |
| 36 | "Example line: tokens label, please output according to this example" |
| 37 | "Return all components and their quantized coordinate indices" |

### A.10.3 TYPE 2: MULTI-PART GROUNDING (FINE TEXT)

| ID | Prompt Template |
|---|---|
| 1 | "Please describe the overall appearance of this point cloud in detail, then introduce each part one by one (with AABB tokens)" |
| 2 | "First give an overall impression, then explain each part in turn with bounding box tokens" |
| 3 | "Please provide an overview of this model, and describe each component with tokens" |
| 4 | "What is the overall shape like? What are the materials and functions of each part?" |
| 5 | "Please first introduce the complete structure, then list parts with tokens + detailed explanations" |
| 6 | "From this point cloud, give an overall description then detail each part with its bounding box" |
| 7 | "Describe the complete object, followed by part-wise details using quantized tokens" |

| ID | Prompt Template |
|---|---|
| 8 | "Provide a holistic view and then list all elements with 6 box tokens and properties" |
| 9 | "Summarize the scene, then output each component in the required token format" |
| 10 | "Give a full description first, then annotate every part with its box tokens and long caption" |
| 11 | "Please first present the overall features, then elaborate on each functional component" |
| 12 | "After summarizing the appearance, list each part item by item (format: tokens description)" |
| 13 | "Give the global appearance, then each part line starts with 6 tokens" |
| 14 | "Present the overall structure and afterwards the detailed attributes of all components" |
| 15 | "Explain the general design; afterwards specify each element with its tokens and features" |
| 16 | "First output an overall description, then write a detailed explanation for each part with tokens" |
| 17 | "Describe holistically, then provide component-wise explanations with bounding-box indices" |
| 18 | "Begin with the object overview; subsequently list parts and their detailed properties" |
| 19 | "Offer a complete summary and then enumerate parts with tokenized boxes" |
| 20 | "Return the overall description and AABB + detailed explanation for each part" |
| 21 | "Finally, please list all components and their features in the specified format" |
| 22 | "Please output in the format of 'overall description tokens description'" |
| 23 | "Provide each part in turn (including token bounding box and function/material description)" |
| 24 | "Provide the overall description followed by every part in the required tokenized box format" |
| 25 | "Please finish by listing each component's six box tokens and an informative sentence" |
| 26 | "Return first the global description, then each element as tokens LONG_DESCRIPTION" |
| 27 | "Include a holistic summary, then annotate each part with its quantized AABB and details" |
| 28 | "Conclude with the part-wise list using bounding-box tokens plus their detailed attributes" |
| 29 | "Output the parts list, each line starting with tokens" |
| 30 | "Please output the description of this object or scene and its parts' BBox information, overall first then parts, format and order cannot be changed" |
| 31 | "End by outputting all parts and their respective detailed features" |
| 32 | "Summary first, then component lines with tokens and descriptions" |
| 33 | "Output strictly in two sections: overview + per-part details" |
| 34 | "After the overview, enumerate every part with its quantized box tokens" |
| 35 | "Overall + parts format example: tokens The left handle is ..." |

Table 12: All-task results on the 400-case unseen benchmark. "Type/Name" follows the template definitions in Table 10. Blank entries indicate that the GT for that task does not contain the corresponding modality.

| Task | Type | Name | IoU | SBERT | SimCSE | BLEU-1 | ROUGE-L | METEOR |
|---|---|---|---|---|---|---|---|---|
| 0 | T0 | Pure box listing | 0.755 | | | | | |
| 1 | T1 | Multi-Part Grounding (Q1) | 0.728 | 55.60 | 54.19 | 35.55 | 35.58 | 18.09 |
| 2 | T2 | Multi-Part Grounding (Q2) | 0.736 | 63.68 | 60.68 | 31.01 | 33.68 | 27.72 |
| 3 | T3 | Single-Part Grounding (Q1) | 0.528 | 73.28 | 71.70 | 36.29 | 38.94 | 33.21 |
| 4 | T4 | Single-Part Grounding (Q2) | 0.443 | | | | | |
| 5 | T5 | Box-to-Text (Q1) | | 57.35 | 56.49 | 38.12 | 38.14 | 19.49 |
| 6 | T6 | Box-to-Text (Q2) | | 64.64 | 61.96 | 31.35 | 33.73 | 28.13 |
| 7 | T7 | Part QA | 0.554 | 78.98 | 84.25 | 40.54 | 42.26 | 34.24 |
| 8 | T8 | Edit—Remove (program) | 0.473 | | | | | |
| 9 | T9 | Edit—Replace (program) | 0.409 | | | | | |
| 10 | T10 | Edit—Add (program) | 0.700 | 80.38 | 79.71 | 47.62 | 51.66 | 46.63 |

## A.10.4 TYPE 3: SINGLE-PART GROUNDING (FROM COARSE TEXT)

| ID | Prompt Template |
|---|---|
| 1 | "Find the {part_name} in this model" |
| 2 | "Locate the {part_name} in this model" |
| 3 | "Point out the {part_name} in this point cloud" |
| 4 | "Mark the {part_name} in this object" |
| 5 | "Where is the {part_name} in this 3D model?" |
| 6 | "Identify the {part_name} in this point cloud" |
| 7 | "Please show all {part_name} in this object" |
| 8 | "Where is the position of {part_name} in this scene?" |
| 9 | "Locate the {part_name} in this model" |
| 10 | "Find the {part_name} in this point cloud" |
| 11 | "Point out the {part_name} in this object" |
| 12 | "Where is the {part_name} in this 3D shape?" |
| 13 | "Mark the {part_name} in this model" |
| 14 | "Show all {part_name} in this object" |
| 15 | "Identify the {part_name} in this point cloud" |
| 16 | "Highlight the position of {part_name}" |

## A.10.5 TYPE 4: SINGLE-PART GROUNDING (FROM FINE TEXT)

| ID | Prompt Template |
|---|---|
| 1 | "Where is the part corresponding to this description: {part_description}" |
| 2 | "Help me locate this part: {part_description}" |
| 3 | "Find the corresponding part based on this description: {part_description}" |
| 4 | "In this point cloud, which part does {part_description} refer to?" |
| 5 | "Mark the position of this part: {part_description}" |
| 6 | "Please provide the bounding box for the part corresponding to this description: {part_description}" |
| 7 | "Find the part that matches this description: {part_description}" |
| 8 | "Locate the component described as: {part_description}" |

| ID | Prompt Template |
|---|---|
| 9 | "Which part is this referring to: {part_description}" |
| 10 | "Mark the boundary of: {part_description}" |
| 11 | "Show the box coordinates for: {part_description}" |
| 12 | "Provide the bounding box for this described element: {part_description}" |
| 13 | "Where exactly is: {part_description}" |
| 14 | "Given this description, locate the corresponding part: {part_description}" |
| 15 | "Locate the part based on this text and provide its AABB: {part_description}" |
| 16 | "Which specific part does this description correspond to? {part_description}" |

### A.10.6  TYPE 5: BOX-TO-TEXT (COARSE)

| ID | Prompt Template |
|---|---|
| 1 | "What is this part?" |
| 2 | "What is this marked area?" |
| 3 | "What is contained in this box?" |
| 4 | "What is this marked portion called?" |
| 5 | "What part is inside this bounding box?" |
| 6 | "What is this part called?" |
| 7 | "Name this highlighted component" |
| 8 | "What is contained in this bounding box?" |
| 9 | "Identify this marked region" |
| 10 | "Give the name of this part" |
| 11 | "What is inside this AABB box?" |
| 12 | "Name this area with one word" |
| 13 | "What's the simple label for this bounded area?" |
| 14 | "What would you call this boxed element?" |
| 15 | "What part does this bounding box point to? Please answer briefly" |
| 16 | "What is this outlined section?" |
| 17 | "Provide the name for this demarcated part" |

### A.10.7  TYPE 6: BOX-TO-TEXT (FINE)

| ID | Prompt Template |
|---|---|
| 1 | "Describe this part in detail" |
| 2 | "What does this area contain? Please explain in detail" |
| 3 | "Please describe the part within this bounding box, including appearance, material and function" |
| 4 | "What is in this box? Please provide detailed information" |
| 5 | "What is the marked portion? Please provide a complete description" |
| 6 | "Describe this part in detail" |
| 7 | "What can you tell me about this highlighted component?" |
| 8 | "Provide a comprehensive description of what's in this box" |

| ID | Prompt Template |
|---|---|
| 9 | "Explain the appearance, material and function of this marked area" |
| 10 | "Give details about this bounded region" |
| 11 | "What are the characteristics of this marked area? Please describe comprehensively" |
| 12 | "Elaborate on the appearance and purpose of this part" |
| 13 | "What is contained in this bounding box? Elaborate on its features" |
| 14 | "Tell me everything about this outlined element" |
| 15 | "What is the material, shape and function of the object in this box?" |
| 16 | "Please characterize this demarcated component thoroughly" |
| 17 | "What's inside this box? Include all relevant details" |

### A.10.8 TYPE 7: PART-AWARE Q&A

This task reuses the questions from the 'QA' field in the raw annotations and replaces textual part references with box tokens in the answer. No new templates are generated for the questions themselves.

### A.10.9 TYPE 8: DELETION PROGRAM

| ID | Prompt Template |
|---|---|
| | *By part name* |
| 1 | "Please remove the {part_name} from this object" |
| 2 | "Get rid of every {part_name}" |
| 3 | "I want to delete the {part_name} here" |
| 4 | "Can you erase all instances of the {part_name}?" |
| 5 | "Show me this model but without the {part_name}" |
| 6 | "Take out the {part_name}" |
| 7 | "The {part_name} needs to be removed" |
| 8 | "Omit the {part_name} from this scene" |
| 9 | "I don't want to see the {part_name} anymore" |
| 10 | "Could you proceed with deleting the {part_name}?" |
| 11 | "Let's see what it looks like if we remove the {part_name}" |
| 12 | "Exclude the {part_name} from the final output" |
| 13 | "The task is to get rid of the {part_name}" |
| 14 | "Wipe out the {part_name} from the 3D model" |
| 15 | "Please filter out the {part_name}" |
| 16 | "Delete the component identified as {part_name}" |
| 17 | "I require the removal of the {part_name}" |
| 18 | "Make the {part_name} disappear" |
| 19 | "This model would be better without the {part_name}" |
| 20 | "Execute the deletion of the {part_name}" |
| | *By part description* |
| 21 | "Please remove this specific part: "part_description"" |
| 22 | "I don't want the component described as "part_description"" |
| 23 | "Delete the part that is "part_description"" |

| ID | Prompt Template |
|----|-----------------|
| 24 | `"Get rid of this particular element:`
`part_description"` |
| 25 | `"Find the part matching part_descriptionänd remove`
`it"` |
| 26 | `"The element characterized by part_descriptionshould`
`be deleted"` |
| 27 | `"Erase the component with this description:`
`part_description"` |
| 28 | `"I want to exclude the part that is part_description"` |
| 29 | `"Locate and then delete this item: part_description"` |
| 30 | `"Take out the part that looks like this:`
`part_description"` |
| 31 | `"The target for deletion is the part described as:`
`part_description"` |
| 32 | `"Can you remove the part with these features:`
`part_description"` |
| 33 | `"Please omit this from the model: part_description"` |
| 34 | `"Based on the description part_description, remove`
`the corresponding part"` |
| 35 | `"I've identified a part to remove: part_description"` |
| 36 | `"Wipe the following item from the scene:`
`part_description"` |
| 37 | `"The part to be erased is: part_description"` |
| 38 | `"Remove the object that fits this profile:`
`part_description"` |
| 39 | `"Please execute a deletion on the component`
`identified as part_description"` |
| 40 | `"Let's remove one specific part: part_description"` |

A.10.10   TYPE 9: MODIFICATION PROGRAM

| ID | Prompt Template |
|----|-----------------|
| 1 | `"Please edit the {part_name} to be {new_description}"` |
| 2 | `"Change the {part_name} into {new_description}"` |
| 3 | `"Replace the {part_name} with this:`
`{new_description}"` |
| 4 | `"I want the {part_name} to look like this:`
`{new_description}"` |
| 5 | `"Modify the {part_name} to become {new_description}"` |
| 6 | `"Update the {part_name} so it is now`
`{new_description}"` |
| 7 | `"Let's alter the {part_name}. It should be`
`{new_description}"` |
| 8 | `"Transform the {part_name} into {new_description}"` |
| 9 | `"Could you make the {part_name} to be`
`{new_description}"` |
| 10 | `"My instruction is to change the {part_name} to`
`{new_description}"` |
| 11 | `"The {part_name} needs an update. Here are the new`
`details: {new_description}"` |
| 12 | `"Let's swap the current {part_name} with a new one:`
`{new_description}"` |
| 13 | `"The {part_name} should be revised to be`
`{new_description}"` |

| ID | Prompt Template |
|----|-----------------|
| 14 | "Please perform an edit on the {part_name}. It should now be {new_description}" |
| 15 | "Adjust the {part_name} to match this description: {new_description}" |

### A.10.11 TYPE 10: ADDITION PROGRAM

| ID | Prompt Template |
|----|-----------------|
| 1 | "Add the {part_name} to this 3D asset." |
| 2 | "Please add a {part_name} to the model." |
| 3 | "Insert the {part_name} component." |
| 4 | "Attach the {part_name} to this object." |
| 5 | "Place the {part_name} on this model." |
| 6 | "Include the {part_name} in this design." |
| 7 | "Incorporate the {part_name} into this structure." |
| 8 | "This model is missing its {part_name}. Please add it." |
| 9 | "Complete this 3D model by adding the {part_name}." |
| 10 | "The {part_name} is missing. Add it back." |
| 11 | "Restore the {part_name} to this object." |
| 12 | "Fill in the missing {part_name}." |
| 13 | "This asset needs a {part_name}. Add it." |
| 14 | "Enhance this model with a {part_name}." |
| 15 | "Improve this design by adding the {part_name}." |
| 16 | "Augment this object with the {part_name}." |
| 17 | "Extend this model to include the {part_name}." |
| 18 | "Could you add the {part_name} to complete this model?" |
| 19 | "I need you to add the {part_name} to this 3D object." |
| 20 | "Would you please attach the {part_name}?" |
| 21 | "Can you help me add the {part_name} component?" |
| 22 | "Mount the {part_name} in the appropriate position." |
| 23 | "Install the {part_name} where it belongs." |
| 24 | "Position the {part_name} correctly on this model." |
| 25 | "Generate and add the {part_name} to this asset." |
| 26 | "Create the {part_name} component for this model." |
| 27 | "Design and attach the {part_name}." |
| 28 | "This looks incomplete without the {part_name}. Add it." |
| 29 | "To make this functional, add the {part_name}." |
| 30 | "The model requires a {part_name} to be complete." |
| *Part-specific templates* | |
| 31 | "Add the head section to complete this figure." |
| 32 | "This model needs its head. Please attach it." |
| 33 | "The top part is missing. Add the head." |
| 34 | "Install the wheels to make this vehicle complete." |
| 35 | "Add wheels for mobility." |
| 36 | "Mount the wheels on this vehicle." |
| 37 | "Install the door to complete the entrance." |
| 38 | "Add a door for access." |
| 39 | "Place the door in the opening." |
| 40 | "Attach the handle for better grip." |
| 41 | "Add the handle component." |
| 42 | "Install the handle mechanism." |

| ID | Prompt Template |
|----|-----------------|
| 43 | `"Add the legs to support this structure."` |
| 44 | `"Attach the leg components."` |
| 45 | `"Install the supporting legs."` |
| 46 | `"Add wings to complete this model."` |
| 47 | `"Attach the wing components."` |
| 48 | `"Install the wings on both sides."` |

