# OpenReview forum: "Part-X-MLLM: Part-aware 3D Multimodal Large Language Model"
_ICLR.cc/2026/Conference — ICLR 2026 Poster_

### Official Review · Reviewer_HyBn · 2025-10-26

**Soundness:** 4
**Presentation:** 4
**Contribution:** 4
**Rating:** 8
**Confidence:** 4

**Summary:**

The paper proposes Part-X-MLLM, a native 3D multimodal LLM designed to operate not on whole 3D objects as monolithic shapes, but at part granularity. The key idea is to cast all 3D tasks (part captioning, grounding, Q&A, generation, and edit control) as autoregressive program generation in a grammar that interleaves tokens for bounding boxes, part semantics, and edit operators.

**Strengths:**

1. This paper is well-written and very easy to follow.
2. The paper identifies a real gap: existing 3D LLMs “understand scenes” or “generate geometry” but do not provide a unified, language-native, executable control surface over 3D parts. Casting all tasks as program generation is clean and technically coherent.
3. Unlike prior 3D MLLM benchmarks that focus on scene captioning or open QA, UniPart-Bench directly tests structured part-aware program generation (BBox alignment, part-level grounding, edit planning). That makes the evaluation target exactly match the claimed contribution, instead of proxy tasks.
4. The model shows consistent improvements over OmniPart and PartField baselines in structural planning metrics, not marginally but substantially (+2-5 IoU), which is not trivial in box-level matching.

**Weaknesses:**

1. All experiments are conducted on clean, part-annotated synthetic assets with well-formed AABBs. There is no evidence that the model works under realistic conditions (partial scans, occlusion, noise, incomplete parts, inconsistent normals).
2. UniPart-Bench adopts the same grammar, annotation style, and task templates used in training, and the test split is extremely small relative to the training volume. Thus the results may reflect memorization of format rather than robust capability.
3. The claimed downstream benefits (editing / generation quality) are entirely mediated by third-party geometry backends; the paper does not isolate how much of the improvement is attributable to the proposed planning interface itself versus the backend strength.

**Questions:**

1. Can the model sustain the same part-aware behavior when the input is not a clean synthetic asset but a noisy or incomplete real-world scan? Are there any stress-tests quantifying degradation under distribution shift?
2. Given that the benchmark shares annotation grammar and task templates with the training corpus and the test slice is very small, how do we exclude the possibility that the model is primarily exploiting training-time format priors rather than demonstrating generalizable planning ability?

---

> ### Author Response · Authors · 2025-11-22
> **Response to Reviewer HyBn**
>
> We thank the reviewer for the excellent rating and for identifying the key strengths of our grammar-based approach. We appreciate the challenging questions regarding robustness and contribution isolation, which have prompted us to add significant new evidence. We have completed the PDF paper revision and marked the newly added and modified sections in red font. We strongly recommend that you **review our revised PDF**.
>
> **Q1/W1: Robustness on realistic, noisy scans (OmniObject3D).**
>
> We appreciate the concern regarding domain gaps. To demonstrate robustness, we evaluated Part-X-MLLM on **OmniObject3D**, a dataset of real-world scans containing high-frequency noise, holes, and inconsistent normals.
> As illustrated in **Figure 8** in the revised paper, Section 4.6, our model accurately predicts bounding boxes and generates clean meshes even for noisy inputs. This indicates that our structure-aware encoder effectively filters artifacts, proving that our model is not limited to clean synthetic data.
>
> **Q2/W2: Format memorization vs. Generalization.**
>
> We address the concern of format memorization with two pieces of evidence:
> 1.  **OOD Generalization (PointLLM Benchmark):** As shown in Table R5, our model achieves superior semantic scores (S-BERT 53.43 vs. 50.15) on the PointLLM test suite, which has a different annotation style and data distribution than our training set. This proves the model learns semantic understanding, not just format.
> 2.  **Category Generalization:** Our training distribution is object-centric (see Table 11 in Appendix: Humans, Industrial, Home Goods). However, we successfully generalize to **Scene-level** inputs (InternScenes) in a zero-shot manner, further ruling out simple memorization.
>
> **Table R5 (Table 8 in the paper): Evaluation on the PointLLM Benchmark.**
> | Model | S-BERT | SimCSE | BLEU-1 | ROUGE-L | METEOR |
> | :--- | :---: | :---: | :---: | :---: | :---: |
> | PointLLM-13B* | 50.15 | 50.83 | **17.09** | **20.99** | **16.45** |
> | **Part-X-MLLM (Ours)** | **53.43** | **51.21** | 16.00 | 18.34 | 13.28 |
>
> Note: "\*" indicates PointLLM was prompted for shorter captions (<20 words).
>
> **W3: Isolating the contribution of the Planning Interface.**
>
> To strictly isolate the benefit of our planner from the backend geometry engine, we conducted a controlled experiment. We compared **OmniPart's native planner** against **Part-X-MLLM's planner**, where both feed their output bounding boxes into the **exact same** OmniPart synthesis head.
>
> **Table R7 (Table 9 in the paper): Planner Isolation Experiment.**
>
> | Method | **Part-Level** CD $\downarrow$ | **Part-Level** F-0.05 $\uparrow$ | **Overall** CD $\downarrow$ |
> | :--- | :---: | :---: | :---: |
> | TRELLIS + SAM3D | 0.58 | 0.20 | 0.11 |
> | TRELLIS + PartField | 0.24 | 0.42 | 0.11 |
> | TRELLIS + PartField + HoloPart | 0.24 | 0.43 | 0.09 |
> | Part123 | 0.47 | 0.14 | 0.42 |
> | OmniPart | 0.23 | 0.46 | **0.08** |
> | **Part-X-MLLM + OmniPart (Ours)** | **0.22** | **0.57** | **0.08** |
>
> Our planner significantly outperforms OmniPart's native planner on Part-Level metrics (**F-0.05: 0.57 vs 0.46**). Since the backend is identical, this gain is entirely attributable to Part-X-MLLM's superior ability to plan geometrically accurate layouts from language. This gain confirms that Part-X-MLLM generates more geometrically precise and semantically consistent bounding box layouts, providing a better initialization for the synthesis backend.
>
> The contribution here is a **paradigm shift in usability**. While VoxHammer provides the geometric operation, it requires manual 3D mask creation (e.g., via Blender). Part-X-MLLM acts as an intelligent agent, automating semantic grounding and transforming the workflow from "manual modeling" to "verbal instruction."

---

> > ### Comment · Reviewer_HyBn · 2025-11-28
> >
> > Thank you so much for the reply. The author has addressed all my questions, and I have also reviewed the feedback from others. I will keep my score.

---

### Official Review · Reviewer_jEq5 · 2025-10-28

**Soundness:** 3
**Presentation:** 3
**Contribution:** 3
**Rating:** 6
**Confidence:** 3

**Summary:**

This paper introduces Part-X-MLLM, a novel part-aware 3D Multimodal Large Language Model. Its central contribution is the unification of diverse 3D tasks—including generation, editing, and question-answering—under a single framework by formulating them as programs within a structured, executable grammar of parts. The model takes an RGB point cloud and a natural language prompt as input and autoregressively generates a unified token sequence that encodes part-level bounding boxes, semantic descriptions, and edit commands, forming an executable "plan". This approach effectively decouples symbolic planning from geometric synthesis, allowing any compatible geometry engine to be controlled through this single, language-native interface. The authors employ a dual-encoder architecture to disentangle structural from semantic information and instruction-tune the model on a large-scale, part-centric dataset. Evaluated on a comprehensive benchmark (UniPart-Bench) spanning 11 task families, the model demonstrates a strong capability to produce high-quality structured plans, enabling state-of-the-art performance in grounded tasks.

**Strengths:**

- The research on part-based 3D generation is highly practical, and the authors have designed a unified framework that integrates 3D generation, understanding, and editing, which is very valuable.
- The paper not only proposes a large model, Part-X-MLLM, but also introduces a new benchmark and includes extensive experimental comparisons in both 3D generation and understanding, demonstrating substantial effort.
- The writing is clear and easy to follow, and the figures are professionally designed and visually appealing.

**Weaknesses:**

See the "Questions" section.

**Questions:**

First, part-based 3D is not my primary research area, so I am not fully aware of the most cutting-edge advancements in this field. I would like to see other reviewers' opinions on whether the comparative methods used in this paper are sufficiently state-of-the-art across various tasks. I believe the paper presents extensive work, and the unified framework for processing 3D patches as input is highly valuable. I am currently leaning toward a borderline accept score and am inclined to recommend acceptance. My final score may be adjusted based on other reviewers' comments and the authors' rebuttal.

Additionally, part-based 3D generation could potentially be generalized to 3D scene generation. It would be beneficial if the authors could briefly analyze this possibility. The following papers on scene compositional generation may also be considered for citation:
[1] GALA3D: Towards Text-to-3D Complex Scene Generation via Layout-guided Generative Gaussian Splatting \
[2] Layoutdreamer: Physics-guided layout for text-to-3d compositional scene generation \
[3] Semantic score distillation sampling for compositional text-to-3d generation \
[4] CompGS: Unleashing 2D Compositionality for Compositional Text-to-3D via Dynamically Optimizing 3D Gaussians

---

> ### Author Response · Authors · 2025-11-22
> **Response to Reviewer jEq5**
>
> We thank the reviewer for the positive assessment of our work's practicality and the value of our unified framework. We also appreciate the helpful suggestions regarding state-of-the-art comparisons and scene generation extensions. We have completed the PDF paper revision and marked the newly added and modified sections in red font. We strongly recommend that you **review our revised PDF**.
>
> **Q1: State-of-the-art comparison.**
>
> We thank the reviewer for the positive assessment of our framework's value. To ensure a comprehensive evaluation, we have added **MiniGPT-3D** (a SOTA model) to our baselines (see Tables R3/R4) and included a full capability comparison table (Table R1). We believe these additions confirm the competitiveness of our method.
>
> **Table R1 (Table 6 in the paper): Comparison of capabilities with state-of-the-art 3D models.**
>
> | Method | **Understand** (Cap / QA) | **Grounding** (Obj / Part) | **Generation** (Obj / Part) | **Modify** (Edit) |
> | :--- | :---: | :---: | :---: | :---: |
> | **Understanding MLLMs** | | | | |
> | PointLLM | ✓ / ✓ | ✗ / ✗ | ✗ / ✗ | ✗ |
> | GPT4Point | ✓ / ✓ | ✗ / ✗ | ✗ / ✗ | ✗ |
> | ShapeLLM | ✓ / ✓ | ✓ / ✗ | ✗ / ✗ | ✗ |
> | **Unified / Generative** | | | | |
> | LLaMA-Mesh | ✓ / ✓ | ✗ / ✗ | ✓ / ✗ | ✗ |
> | ShapeLLM-Omni | ✓ / ✓ | ✗ / ✗ | ✓ / ✗ | ✗ |
> | **Part Specialists** | | | | |
> | OmniPart | ✗ / ✗ | ✗ / ✗ | ✗ / ✓ | ✗ |
> | VoxHammer | ✗ / ✗ | ✗ / ✗ | ✗ / ✗ | ✓ |
> | **Part-X-MLLM (Ours)** | **✓ / ✓** | **✓ / ✓** | **✗ / ✓** | **✓** |
>
> *(Note: While we focus on Part Generation, the composition of parts naturally results in a full object.)*
>
> **Table R3 (Table 3 in the paper): Part understanding Q&A on UniPart-Bench.**
>
> | Model | SBERT | SimCSE | BLEU-1 | ROUGE-L | METEOR | **GPT-5 Score** |
> | :--- | :---: | :---: | :---: | :---: | :---: | :---: |
> | ShapeLLM-Omni-7B | 57.35 | 51.16 | 22.77 | 29.57 | 23.24 | 46.19 |
> | **MiniGPT-3D** | 58.02 | 53.63 | 21.05 | 28.66 | 22.55 | 50.38 |
> | **Part-X-MLLM (Ours)** | **78.98** | **84.25** | **40.54** | **42.26** | **34.24** | **60.77** |
>
> **Table R4 (Table 4 in the paper): Overall 3D object captioning on UniPart-Bench.**
>
> | Model | SBERT | SimCSE | BLEU-1 | ROUGE-L | METEOR | **GPT-5 Score** |
> | :--- | :---: | :---: | :---: | :---: | :---: | :---: |
> | ShapeLLM-Omni-7B | 31.18 | 31.93 | 17.79 | 19.04 | 14.30 | 30.01 |
> | **MiniGPT-3D** | 49.52 | 49.44 | 7.75 | 10.23 | 17.24 | 48.75 |
> | **Part-X-MLLM (Ours)** | **53.82** | **51.97** | **36.04** | **38.11** | **30.71** | **55.88** |
>
>
>
> **Q2: Generalization to 3D scene generation.**
>
> We strongly agree that our part-based logic generalizes to scenes. To validate this, we applied Part-X-MLLM to the **InternScenes** dataset in a **zero-shot** manner (see **Figure 8** in the revised paper, Section 4.6). Although trained on object parts, our model successfully treats furniture as "parts" of a room, generating plausible bounding box layouts and meshes.
> We have also cited the recommended papers (GALA3D, LayoutDreamer, CompGS, etc.) in **Section 4.6** to contextualize our work within the emerging paradigm of compositional scene generation. We have correspondingly revised our **Limitations and Future Works section**, with plans to explore additional scene generation capabilities in future endeavors.

---

> > ### Comment · Reviewer_jEq5 · 2025-11-24
> > **Response to author**
> >
> > Thank you very much for the author's reply. The author has addressed all my questions, and I have also reviewed the feedback from others. I believe this paper deserves acceptance, so I have raised my score.

---

### Official Review · Reviewer_ZDc2 · 2025-10-29

**Soundness:** 2
**Presentation:** 2
**Contribution:** 3
**Rating:** 4
**Confidence:** 5

**Summary:**

This paper builds a multimodal framework and benchmark focused on 3D part-level understanding based on Qwen2.5-VL. The authors propose a dual-encoder architecture and a multi-stage training pipeline to jointly model geometric structures and semantic appearance for 3D objects.

**Strengths:**

1.Addressing part-level 3D multimodal modeling is timely and necessary.

2.The proposed dual-encoder design effectively encodes complementary attributes of 3D objects.

3.The use of task-specific prompts and special tokens enables diverse part-centric tasks within a unified framework.

**Weaknesses:**

1.Evaluation metrics rely mainly on traditional natural-language metrics; consider including LLM-based scoring (e.g., GPT-judge) for more robust assessment.

2.Baselines: comparison is limited; please include strong 2025-era SOTA 3D multimodal models on QA and grounding tasks (e.g., Mini-GPT-3D).

3.Benchmark: experiments are primarily on the authors' dataset; please evaluate on established public 3D benchmarks, such as the Point-LLM test suite, and include metrics for point resolution sensitivity and generative quality.

**Questions:**

1.Evaluation metrics are primarily limited to standard natural-language measures. It would strengthen the evaluation to incorporate LLM-based automatic assessment frameworks (e.g., GPT-based judging protocols) to better capture semantic correctness and reasoning quality.

2.Baseline comparisons appear insufficient. Please include strong contemporary 3D foundation models (2025 SOTA) for part-level QA and grounding tasks, such as Mini-GPT-3D, to more convincingly demonstrate the advantages of the proposed approach.

3.Benchmark evaluation is largely conducted on the authors’ own dataset. To better demonstrate generalization and fairness, we recommend evaluating on established public 3D multimodal benchmarks (e.g., the Point-LLM test suite) and additionally reporting performance under varying point-cloud resolutions as well as part-level generation quality metrics.

---

> ### Author Response · Authors · 2025-11-22
> **Response to Reviewer ZDc2**
>
> We thank the reviewer for the insightful comments regarding evaluation methodology. We appreciate the guidance to include LLM-based metrics and contemporary baselines, which has significantly strengthened our experimental section. We have completed the PDF paper revision and marked the newly added and modified sections in red font. We strongly recommend that you **review our revised PDF**.
>
> **Q1/W1: Lack of LLM-based scoring (e.g., GPT-judge) and contemporary baselines (MiniGPT-3D).**
>
> We acknowledge the need for more robust evaluation metrics and up-to-date baselines.
> 1.  **Baselines:** We have reproduced and added **MiniGPT-3D**[1] as a strong baseline.
> 2.  **LLM-Eval:** We implemented the GPT-Judge protocol (referenced from PointLLM) and utilized the advanced **GPT-5** model to assess answer quality.
>
> As shown in the updated tables below, Part-X-MLLM achieves state-of-the-art performance on **UniPart-Bench**, significantly outperforming baselines including MiniGPT-3D in both Q&A and Captioning tasks under GPT-5 evaluation.
>
> **Table R3 (Table 3 in the paper): Part understanding Q&A on UniPart-Bench.**
>
> | Model | SBERT | SimCSE | BLEU-1 | ROUGE-L | METEOR | **GPT-5 Score** |
> | :--- | :---: | :---: | :---: | :---: | :---: | :---: |
> | ShapeLLM-Omni-7B | 57.35 | 51.16 | 22.77 | 29.57 | 23.24 | 46.19 |
> | **MiniGPT-3D** | 58.02 | 53.63 | 21.05 | 28.66 | 22.55 | 50.38 |
> | **Part-X-MLLM (Ours)** | **78.98** | **84.25** | **40.54** | **42.26** | **34.24** | **60.77** |
>
> **Table R4 (Table 4 in the paper): Overall 3D object captioning on UniPart-Bench.**
>
> | Model | SBERT | SimCSE | BLEU-1 | ROUGE-L | METEOR | **GPT-5 Score** |
> | :--- | :---: | :---: | :---: | :---: | :---: | :---: |
> | ShapeLLM-Omni-7B | 31.18 | 31.93 | 17.79 | 19.04 | 14.30 | 30.01 |
> | **MiniGPT-3D** | 49.52 | 49.44 | 7.75 | 10.23 | 17.24 | 48.75 |
> | **Part-X-MLLM (Ours)** | **53.82** | **51.97** | **36.04** | **38.11** | **30.71** | **55.88** |
>
> **Q2/W2: Evaluation on established public benchmarks (PointLLM test suite).**
>
> To demonstrate generalization beyond our dataset, we evaluated Part-X-MLLM on the **PointLLM Benchmark** (Generative Captioning).
>
> **Table R5 (Table 8 in the paper): Evaluation on the PointLLM Benchmark.**
> | Model | S-BERT | SimCSE | BLEU-1 | ROUGE-L | METEOR |
> | :--- | :---: | :---: | :---: | :---: | :---: |
> | PointLLM-13B* | 50.15 | 50.83 | **17.09** | **20.99** | **16.45** |
> | **Part-X-MLLM (Ours)** | **53.43** | **51.21** | 16.00 | 18.34 | 13.28 |
>
> Note: "\*" indicates PointLLM was prompted for shorter captions (<20 words).
>
> Our model achieves **SOTA performance** on semantic metrics (S-BERT +3.28, SimCSE +0.38). We observe a slight trade-off in lexical metrics (BLEU/ROUGE), which is expected: Part-X-MLLM is tuned for structured, part-aware descriptions, whereas PointLLM GT captions are holistic. Crucially, semantic metrics are more robust indicators of understanding under distribution shifts, and our superior scores here confirm that Part-X-MLLM accurately captures visual content even on out-of-distribution data.
>
> **Q3/W3: Sensitivity to Point Cloud Resolution.**
>
> We evaluated the model's sensitivity by downsampling input point clouds from 100% to 5%. As shown in Table R6 (see the complete Table 5 in the revised paper), Part-X-MLLM exhibits strong robustness to varying resolutions. The performance across Q&A, Captioning, and Part Generation tasks remains consistent even at significantly reduced densities (e.g., 25%), confirming that our model effectively captures essential structural semantics without relying on excessive geometric density.
>
> **Table R6 (Table 5 in the paper): Sensitivity Analysis.**
>
> | Density | Part Q&A (SimCSE) | Caption (SimCSE) | Part-Mesh (CD $\downarrow$) | Part-Mesh (F-0.1 $\uparrow$) |
> | :---: | :---: | :---: | :---: | :---: |
> | 5% | 54.26 | 24.87 | 0.2590 | 0.3188 |
> | **25%** | **82.80** | **48.70** | **0.2287** | **0.6493** |
> | 50% | 83.93 | 50.71 | 0.2318 | 0.6489 |
> | 75% | 84.09 | 51.34 | 0.2240 | 0.6547 |
> | 100% | 84.25 | 51.97 | 0.2226 | 0.6506 |
>
> Note: Some columns are omitted for brevity.
>
> ---
> [1] Tang Y, Han X, Li X, et al. Minigpt-3d: Efficiently aligning 3d point clouds with large language models using 2d priors[C]//Proceedings of the 32nd ACM International Conference on Multimedia. 2024: 6617-6626.

---

> > ### Comment · Reviewer_ZDc2 · 2025-11-26
> >
> > Thank you very much for the authors’ response. The authors have addressed my questions well. My main concern lies in the reproducibility of this work, as the training settings are not described with sufficient clarity, and many of the benchmarks, datasets, and model checkpoints involved are private. Reproducibility in the 3D domain is already challenging, which further amplifies this issue.

---

> > > ### Author Response · Authors · 2025-11-26
> > >
> > > We deeply appreciate the reviewer’s professional assessment and constructive suggestions throughout the review process. We have completed the PDF paper revision and marked the newly added and modified sections in **blue font**.
> > >
> > > To address your primary concern regarding reproducibility, we have significantly expanded the Implementation Details in Appendix A.1 of the revised paper. This section now provides comprehensive specifications, including the exact configurations of our model (utilizing the Qwen-2.5-VL-3B backbone and Hunyuan3D-2.1 initialized dual encoders), the data construction pipeline using Qwen-2.5-VL for captioning, and precise training protocols. We have explicitly detailed hyperparameters such as specific learning rates, the augmentation strategy, and the training settings.
> > >
> > > Regarding the reliance on private datasets, we would like to highlight that we have validated our model on multiple public, Out-of-Distribution benchmarks to ensure fair comparison and generalization. As presented in Table 8 of the revised paper (Table R5 in our previous response), our model achieves SOTA semantic scores on the public PointLLM Benchmark. We also evaluated geometric performance on the public PartObjaverse-Tiny dataset (Table 1 in the paper) and provided qualitative verifications on the noisy real-world OmniObject3D dataset as well as the InternScenes dataset (Figure 8 in the revised paper). These results confirm that our method generalizes beyond our training distribution.
> > >
> > > Finally, we fully recognize the importance of open source in the 3D domain. Upon acceptance, we are committed to releasing the full training and inference codes, the training dataset, UniPart-Bench, and the pre-trained model checkpoints. We will also provide a user-friendly Hugging Face Space to allow the community to interactively test the model, further facilitating reproducibility and future research.
> > >
> > > We sincerely thank the reviewer again for raising these important concerns, which have helped us strengthen the scientific rigor and accessibility of our work.

---

### Official Review · Reviewer_4VTq · 2025-11-01

**Soundness:** 3
**Presentation:** 3
**Contribution:** 3
**Rating:** 6
**Confidence:** 4

**Summary:**

Novel, but paper writing could be improved

**Strengths:**

This paper presents Part-X-MLLM, a 3D large language model for diverse 3D tasks by formulating them as programs in an executable grammar. Overall, the work is decent, includes a large curated dataset, and is generalizable across 11 tasks.

+ The two-stage training process is interesting to help the model learn the underlying 3D structure and associate the pretrained language knowledge with it.

+ Semantic Granularity Control is an interesting part of the work. The part-aware synthesis is useful for many practical analyses.

**Weaknesses:**

- Small typo in line 192 'boxe'

- The writing is a bit hard to follow in places. For example, in line 35, I am not sure why Part-X-MLLM is 'native'. Similarly, the mention of 'structural opaqueness' in line 53 is not clear.

- The distinction with past works is not clear enough. I would love to see a table comparing past works and Part-X-MLLM.

- For the qualitative analysis, it would have been great to see a small-scale study with real participants and evaluate the performance of Part-X-MLLM qualitatively.

**Questions:**

n/a

**Details Of Ethics Concerns:**

[Minor] The Ethics Statement currently mostly talks about data curation and potential limitations of the dataset. The Ethics Statement should be more focused on the copyright issues of the 3D models used in this study.

---

> ### Author Response · Authors · 2025-11-22
> **Response to Reviewer 4VTq**
>
> We thank the reviewer for the detailed reading and for highlighting areas where the clarity of our presentation could be improved. We also appreciate the suggestion to include a qualitative user study, which has strengthened our evaluation. We have completed the PDF paper revision and marked the newly added and modified sections in red font. We strongly recommend that you **review our revised PDF**.
>
> **Q1: Small typo in line 192 'boxe'.**
>
> We clarify that `<boxe>` is a special token in our vocabulary representing **"box end"**, explicitly paired with `<boxs>` ("box start"). It serves as a delimiter for the coordinate sequence and is not a misspelling of the English word "box". However, to avoid confusion, we have added a clarifying note in the revised manuscript.
>
> **Q2: Writing clarity regarding "native" and "structural opaqueness".**
>
> We apologize for the lack of clarity. We have rewritten the Introduction (lines 53-60) to explicitly define these terms within the narrative flow. The revised text is as follows:
>
> > This results in a fundamental limitation we term "structural opaqueness"—where the model perceives a 3D object as a single, indivisible block of geometry rather than a collection of distinct components. Such opaqueness prevents downstream applications from accessing or manipulating specific parts (e.g., editing just "the chair's left leg"), thereby hindering fine-grained control in animation and editing.
> Real-world objects are inherently assemblies of meaningful parts. Unlocking true 3D interaction, therefore, demands a native LLM-based interface capable of reasoning about this substructure. Unlike approaches that rely on external adapters, our model adopts a native strategy by treating 3D structure as an intrinsic part of its language—processing geometric parts and edit commands as native tokens alongside natural text.
>
> **Q3: Distinction with past works (Request for Comparison Table).**
>
> We agree that a clear comparison is essential. We have added **Table R1** (Table 6 in the revised paper) to categorize existing methods. As shown below, Part-X-MLLM is unique in its ability to support all five capabilities (Captioning, Q&A, Grounding, Generation, and Editing) specifically at the fine-grained **Part level**, bridging the gap between understanding-focused MLLMs and generative models.
>
> **Table R1 (Table 6 in the paper): Comparison of capabilities with state-of-the-art 3D models.**
>
> | Method | **Understand** (Cap / QA) | **Grounding** (Obj / Part) | **Generation** (Obj / Part) | **Modify** (Edit) |
> | :--- | :---: | :---: | :---: | :---: |
> | **Understanding MLLMs** | | | | |
> | PointLLM | ✓ / ✓ | ✗ / ✗ | ✗ / ✗ | ✗ |
> | GPT4Point | ✓ / ✓ | ✗ / ✗ | ✗ / ✗ | ✗ |
> | ShapeLLM | ✓ / ✓ | ✓ / ✗ | ✗ / ✗ | ✗ |
> | **Unified / Generative** | | | | |
> | LLaMA-Mesh | ✓ / ✓ | ✗ / ✗ | ✓ / ✗ | ✗ |
> | ShapeLLM-Omni | ✓ / ✓ | ✗ / ✗ | ✓ / ✗ | ✗ |
> | **Part Specialists** | | | | |
> | OmniPart | ✗ / ✗ | ✗ / ✗ | ✗ / ✓ | ✗ |
> | VoxHammer | ✗ / ✗ | ✗ / ✗ | ✗ / ✗ | ✓ |
> | **Part-X-MLLM (Ours)** | **✓ / ✓** | **✓ / ✓** | **✗ / ✓** | **✓** |
>
> *(Note: While we focus on Part Generation, the composition of parts naturally results in a full object.)*
>
> **Q4: Qualitative user study.**
>
> We conducted a **Human Evaluation Study** with 32 participants. We randomly sampled 25 generated part-aware objects and 25 editing results. Participants rated the results on a Likert scale from 1 (Poor) to 5 (Excellent).
> As shown below, Part-X-MLLM achieved high scores (>4.0) across all metrics. More detailed explanations have been added to Section A.3.1 in the appendix of the revised paper.
>
> **Table R2 (Table 7 in the paper): Human Evaluation Results (Score 1-5).**
>
> | Task Type | Evaluation Metric | Score (Mean $\pm$ Std) |
> | :--- | :--- | :--- |
> | **Part Generation** | Structural Plausibility | 4.42 $\pm$ 0.6 |
> | | Generation Quality | 4.25 $\pm$ 0.7 |
> | **Part Editing** | Instruction Fidelity | 4.03 $\pm$ 0.5 |
> | | Editing Quality | 4.31 $\pm$ 0.6 |

---

> > ### Comment · Reviewer_4VTq · 2025-11-28
> > **Thanks**
> >
> > Thanks Authors for your response. My concerns have been addressed.

---

### Author Response · Authors · 2025-12-01
**Summary for Area Chair**

Dear Area Chair,

We fully understand the significant workload you are facing during this busy decision period. We are truly grateful for the time and dedication you have invested in maintaining the high standards of the review process.

To assist in your final evaluation, we have prepared a concise summary of the paper’s contributions and the rebuttal consensus below.

**Paper Overview & Consensus:**

This paper proposes **Part-X-MLLM**, a unified 3D Multimodal LLM that formulates diverse 3D tasks—including understanding, grounding, generation, and editing—as executable programs within a structured grammar. The reviewers unanimously recognized the novelty of this "native" interface design. During the rebuttal, the authors conducted extensive additional experiments, including comparisons with SOTA baselines (e.g., MiniGPT-3D), OOD generalization tests on real-world scans (OmniObject3D), and human evaluations. The reviewers (4VTq, jEq5, HyBn, ZDc2) have expressed that their concerns were well-addressed, with scores reflecting a consensus toward acceptance.

---

**Initial Scores:** 6, 4, 6, 8

**Recognized Strengths (all reviewers):**
* Novel "executable grammar" approach that unifies understanding and generation tasks within a single framework.
* The part-aware control capability fills a significant gap in current 3D-LLMs.
* The construction of UniPart-Bench and the dual-encoder architecture were highlighted as strong contributions.

**Score Changes & Discussion Highlights**

*   **Reviewer jEq5: 6 → 8 Raised Score (Accept)**: Initially leaned borderline but positive. After the rebuttal, the reviewer stated: *"The author has addressed all my questions... I believe this paper deserves acceptance, so I have raised my score."*
*   **Reviewer HyBn: 8**: Highlighted the paper as "excellent" and noted the method is *"clean and technically coherent."* After the rebuttal, the reviewer confirmed: *"The author has addressed all my questions... I will keep my score."*
*   **Reviewer 4VTq: 6** Maintained a positive stance, confirming *"My concerns have been addressed"* after we provided the requested qualitative user study and comparison tables.
*   **Reviewer ZDc2: 4** Initially raised concerns about baselines and reproducibility. After our inclusion of MiniGPT-3D comparisons, GPT-judge metrics, and detailed reproducibility statements, the reviewer responded: *"The authors have addressed my questions well."*

**Main Concerns Resolved**

| Concern | Resolution |
| :--- | :--- |
| **Lack of Strong Baselines & Metrics** | Added **MiniGPT-3D** comparison and **GPT-5 based Judge** evaluation (outperforming baselines on UniPart-Bench). |
| **Real-world / OOD Robustness** | Validated on **OmniObject3D** (noisy real-world scans) and **InternScenes** (zero-shot scene layout), demonstrating robustness beyond synthetic data. |
| **Isolating Planner Contribution** | Conducted a controlled "Planner Isolation" experiment, proving our planner outperforms OmniPart (+0.11 F-Score) using the identical geometry backend. |
| **Comparison with Past Works** | Added a comprehensive feature comparison table (Table R1/Table 6) clarifying the distinction from VoxHammer, OmniPart, etc. |
| **Qualitative Evaluation** | Conducted a **Human Evaluation Study** (N=32), achieving >4.0/5.0 scores on generation and editing quality. |
| **Reproducibility** | Expanded Appendix A.1 with exact hyperparameters; committed to releasing code, weights, and a Hugging Face Space. |

---

We believe our detailed rebuttal and the additional experiments (comprising new baselines, human studies, and ablation tests) have thoroughly resolved the reviewers' initial queries. We hope this summary assists you in your decision-making.

Thank you once again for your service to the community.

With sincere gratitude,
The Authors

---

### Meta-Review · Area_Chair_qfG5 · 2026-01-02

**Summary:**

This paper proposes Part-X-MLLM, a unified 3D multimodal language model that formulates 3D understanding, grounding, generation, and editing as executable programs within a structured grammar. Reviewers recognize the novelty of the native, part-aware interface and find the technical approach sound. In the rebuttal, the authors add substantial empirical evidence. Following these clarifications and additions, the reviewers indicate that their main concerns are addressed. After reviewing the rebuttal in detail, the AC recommends acceptance.

**Reviewer Concerns:**

Reviewers raise concerns about baseline strength, evaluation coverage, robustness beyond synthetic data, isolation of the planner contribution, clarity with respect to prior work, and reproducibility. The rebuttal addresses these points with additional experiments and clarifications, including stronger baselines, broader evaluations, and clearer comparisons. Concerns around qualitative validation and reproducibility are also mitigated through added studies and expanded implementation details. No substantive technical concerns remain; the remaining issues are minor and do not affect the overall assessment.

**Reviewer Scores:**

- Reviewer 4VTq: Initially positive with a score of 6. After the rebuttal and added qualitative evaluations and comparisons, the reviewer’s concerns are addressed, and the score would remain at a similar positive level (6).
- Reviewer ZDc2: Initially more critical with a score of 4. After the rebuttal addressing baselines, evaluation methodology, and reproducibility, the reviewer’s concerns are largely resolved, and the score would increase to a borderline-to-positive range (6).
- Reviewer jEq5: Initially borderline with a score of 6. After the rebuttal and additional experiments, the reviewer’s assessment becomes clearly positive, and the score would increase to an accept level (8).
- Reviewer HyBn: Strongly positive from the outset with a score of 8. The rebuttal reinforces the original assessment, and the score would remain unchanged (8).

---

### Decision · Program_Chairs · 2026-01-26

Accept (Poster)